# Astrocytes mediate two forms of spike timing-dependent depression at entorhinal cortex-hippocampal synapses

Irene Martínez-Gallego[†], Heriberto Coatl-Cuaya[†], Antonio Rodriguez-Moreno*

Laboratory of Cellular Neuroscience and Plasticity, Department of Physiology, Anatomy and Cell Biology, Universidad Pablo de Olavide, Sevilla, Spain

## eLife Assessment

This **valuable** study reports the existence of specific spike-timing dependent synaptic plasticity processes at two excitatory synapses of the dentate gyrus granule cells. These synapses link the entorhinal cortex and the dentate gyrus but via different circuits. With state-of-the-art patch-clamp electrophysiological analysis, the authors provide **convincing** information on the molecular mechanisms underlying these 2 forms of synaptic plasticity showing a critical role for astrocytes in both alongside some features distinctive to each pathway. These results will be of interest to neuroscientists as they uncover detailed plasticity mechanisms involving the hippocampus.

*For correspondence:
arodmor@upo.es

†These authors contributed equally to this work

Competing interest: The authors declare that no competing interests exist.

## Abstract

The entorhinal cortex (EC) connects to the hippocampus sending different information from cortical areas that is first processed at the dentate gyrus (DG) including spatial, limbic and sensory information. Excitatory afferents from lateral (LPP) and medial (MPP) perforant pathways of the EC connecting to granule cells of the DG play a role in memory encoding and information processing and are deeply affected in humans suffering Alzheimer's disease and temporal lobe epilepsy, contributing to the dysfunctions found in these pathologies. The plasticity of these synapses is not well known yet, as are not known the forms of long-term depression (LTD) existing at those connections. We investigated whether spike timing-dependent long-term depression (t-LTD) exists at these two different EC-DG synaptic connections in mice, and whether they have different action mechanisms. We have found two different forms of t-LTD, at LPP- and MPP-GC synapses and characterised their cellular and intracellular mechanistic requirements. We found that both forms of t-LTD are expressed presynaptically and that whereas t-LTD at LPP-GC synapses does not require NMDAR, t-LTD at MPP-GC synapses requires ionotropic NMDAR containing GluN2A subunits. The two forms of t-LTD require different group I mGluR, mGluR5 LPP-GC synapses and mGluR1 MPP-GC synapses. In addition, both forms of t-LTD require postsynaptic calcium, eCB synthesis, $CB_1R$, astrocyte activity, and glutamate released by astrocytes. Thus, we discovered two novel forms of t-LTD that require astrocytes at EC-GC synapses.

## Introduction

The entorhinal cortex (EC) conveys spatial, limbic and sensory information to the hippocampus, a structure responsible for important activities in the brain including learning, memory and spatial information coding. Excitatory projections from the EC to the dentate gyrus (DG) play a role in memory encoding, and they are affected in humans suffering from Alzheimer Disease (AD) and temporal lobe epilepsy, contributing critically to these pathologies (*Paré et al., 1992*; *Llorens-Martín et al., 2014*). In rodents, evidence suggests that the lateral perforant pathway (LPP) conveys more sensory related

information, whereas the medial perforant pathway (MPP) conveys information related to spatial location and limbic signals that are involved in attention and motivation. Evidence suggests that LPP- and MPP-GC synapses have distinct electrophysiological properties, these differences potentially affecting the functional processing of information (*Krueppel et al., 2011*; *Petersen et al., 2013*; *Kim et al., 2018*). Indeed, these functional differences probably reflect their distinct inputs, as they contact the outer (LPP) or middle (MPP) third of the molecular layer of the DG (*Paré et al., 1992*; *Burgess et al., 2002*). Significantly, MPP-GC synapses are associated with a higher probability of release (*Min et al., 1998*) and the existence and/or activity of presynaptic NMDARs, which differs between the two types of fibres (*Savtchouk et al., 2019*). Interestingly, LPP and MPP synapses appear to influence pathological changes distinctly, with the former showing changes in release probability in models of epilepsy (*Scimemi et al., 2006*), preferential β-amyloid deposition (*Reilly et al., 2003*), earlier susceptibility to AD (*Khan et al., 2014*) and reduced plasticity with age (*Froc et al., 2003*). The mechanistic bases of these differences are virtually unknown and they pose a challenge when attempting to understand circuit-specific biological computations, as well as the susceptibility to pathological insults. In addition, the forms of plasticity at these synapses remain unclear and the limited studies into plasticity available did not distinguish between these two pathways.

Plasticity drives the organization of cortical maps during development, and it underlies learning and memory (reviewed in *Malenka and Bear, 2004*; *Mateos-Aparicio and Rodríguez-Moreno, 2019*). The most extensively studied forms of plasticity are long-term potentiation (LTP) and long-term depression (LTD) of synaptic transmission. Spike timing-dependent plasticity (STDP) is a Hebbian form of long-term synaptic plasticity found in all known nervous systems, from insects to humans. Indeed, STDP appears to be a synaptic mechanism that underlies circuit remodelling during development, and it is thought to be involved in learning and memory (*Dan and Poo, 2004*; *Feldman, 2012*). In STDP, the relative order and millisecond timing of pre- and postsynaptic action potentials (spikes/APs) determine the direction and magnitude of the synaptic changes. Thus, spike timing-dependent LTP (t-LTP) occurs when a postsynaptic spike follows a presynaptic spike within ms, whereas spike timing-dependent LTD (t-LTD) is induced when this order is reversed (*Feldman, 2012*). In addition, in vivo studies have suggested modifications to LTP and LTD after mice undertake behavioural tasks (*Kemp and Manahan-Vaughan, 2004*; *Clarke et al., 2010*). However, it is surprising that while plasticity of EC-DG granule cell (GC) synapses has been studied using classical protocols (*Froc et al., 2003*; *Harney et al., 2006*; *Hayashi et al., 2014*; *Savtchouk et al., 2019*), no information exists as to whether t-LTP or t-LTD can be induced at these synapses.

Here we studied t-LTD at PP-GC synapses, differentiating the lateral and medial pathways, determining whether these individual synapses are subject to t-LTD and the mechanisms involved. As a result, we identified two different forms of presynaptic t-LTD at LPP- and MPP-GC synapses. Unlike LPP-GC synapses, the t-LTD at MPP-GC synapses requires ionotropic NMDARs that contain GluN2A subunits, and the t-LTD at each type of synapse require different group I mGluRs, with LPP-GC synapses dependent on mGluR5 and MPP-GC t-LTD requiring mGluR1. In addition, we found that both forms of t-LTD require postsynaptic calcium, endocannabinoid (eCB) synthesis, $CB_1$ receptor ($CB_1R$) signalling, astrocytes and glutamate. Hence, we describe here two novel forms of presynaptic t-LTD in the brain at EC-GC synapses with different action mechanisms, probably implicated in the different aspects of information processing.

## Results

### Pairing presynaptic activity with single postsynaptic action potentials at low frequency can induce t-LTD at mouse lateral and medial perforant pathway-dentate gyrus granule cell synapses

We initially determined whether spike timing-dependent LTD can be induced at perforant pathway-DG GC synapses by pairing presynaptic stimulation with single postsynaptic spikes at low frequency (0.2 Hz) in slices from P13-21 mice. We used a post-pre protocol, pairing a single postsynaptic spike followed 18ms later by presynaptic stimulation 100 times, as described previously at cortical (L4-L2/3) and hippocampal (SC-CA1) synapses (*Andrade-Talavera et al., 2016*; *Martínez-Gallego et al., 2022a*). We monitored the EPSPs evoked by extracellular stimulation of the LPP- and MPP-GC synapses, which make synaptic contacts with the distal and medial third of the dendritic arbour of hippocampal DG

GCs, respectively (*Petersen et al., 2013*; *Savtchouk et al., 2019*). EPSPs were recorded from GCs using the current clamp mode of the whole-cell patch-clamp configuration (*Figure 1a and k*). A post-before-pre pairing protocol induced robust t-LTD at both LPP-GC (63 ± 5 %, n=14: *Figure 1b and c*) and MPP-GC synapses (59 ± 5 %, n=14: *Figure 1l and m*), while unpaired control pathways remained unaffected (103 ± 4% and 105 ± 7%, respectively: *Figure 1b, c, l and m*). We repeated the experiments at LPP-GC and MPP-GC synapses in the presence of GABA$_A$ and GABA$_B$ receptor blockers (bicuculline, 10 µM and SCH50911, 20 µM, respectively) to determine whether they have a role in this form of t-LTD. In these experimental conditions, t-LTD was not affected at these synapses, indicating that these receptors are not necessary for t-LTD induction (LPP-GC: Control t-LTD: 67 ± 6%, n=6; Bicuculline + SCH50911: 69 ± 11%, n=6; *Figure 1—figure supplement 1*). MPP-GC synapses: Control t-LTD: 65 ± 8%, n=6; Bicuculline + SCH50911: 75 ± 9%, n=6; *Figure 1—figure supplement 1*. Thus, we performed the rest of experiments in the absence of any GABA receptor blocker.

## Presynaptic expression of t-LTD at lateral and medial perforant pathway-dentate gyrus granule cell synapses

We next determined the site where this t-LTD was expressed through several approaches. As we observed failures in synaptic transmission at both types of synapses, we analysed these and found that the number of failures increased consistently after t-LTD at both LPP-GC (from 43 ± 7% at baseline to 76 ± 6% after t-LTD, n=8: *Figure 1d*) and MPP-GC synapses (from 40 ± 5% at baseline to 59 ± 4% after t-LTD, n=6: *Figure 1n*), suggesting a presynaptic mechanism. We then analysed the PPR at baseline and 30 min after the pairing protocol, identifying a significant increase in the PPR after t-LTD at both LPP-GC (from 1.16±0.11 at baseline to 1.67±0.19 after t-LTD, n=14: *Figure 1e*) and MPP-GC synapses (from 1.17±0.11 at baseline to 1.81 ± 0.15% after t-LTD, n=14: *Figure 1o*), again suggesting a presynaptic expression of these two forms of t-LTD. Third, we estimated the noise-subtracted CV of the synaptic responses before and after t-LTD induction. A plot of CV$^{-2}$ versus the change in the mean evoked EPSP slope (M) before and after t-LTD mainly yielded points below the diagonal line at LPP-GC and MPP-GC synapses (LPP-GC: Mean = 0.607 ± 0.054 vs 1/CV$^2$=0.439 ± 0.096, R$^2$=0.337; n=14) and MPP-GC (Mean = 0.596 ± 0.056 vs 1/CV$^2$=0.461 ± 0.090, R$^2$=0.168; n=13). *Figure 1f and p*, consistent with a modification of the release parameters (*Rodríguez-Moreno and Paulsen, 2008*; see *Brock et al., 2020* for review). Finally, we recorded and analysed the miniature responses (mEPSP). We performed the experiments in the presence of tetrodotoxin (TTX, 300 nM) to avoid action potentials, before and after t-LTD, adding TTX at the baseline (we added TTX and started to record mEPSP after10 min. of TTX in the bath for additional 10 min) and washing it out for 30 min before performing the t-LTD experiment (we recorded EPSP for 10 min as a baseline after 30 min TTX washout, then applied the t-LTD protocol and then recorded EPSP for additional 30 min), and then added TTX again after t-LTD to record mEPSP for other 10 min (*Falcón-Moya et al., 2020*). This experimental approach serves to determine whether the frequency and amplitude of mEPSP change because of the induction of t-LTD. Under these conditions, a t-LTD similar to previous experiments was evident at both types of synapse (LPP-GC synapses 55 ± 11%, n=6; MPP-GC synapses 67 ± 8%, n=7), and we found that the frequency of mEPSPs decreased at both types of synapse after t-LTD induction: LPP-GC - baseline 1.4±0.13 Hz, after t-LTD induction 0.7±0.09 Hz, n=6 (*Figure 1g and h*); and MPP-GC - baseline 1.2±0.1 H, after t-LTD induction 0.7±0.07 Hz, n=8 (*Figure 1q and r*). No effect on mEPSP amplitude was evident at any of these synapses: LPP-GC - baseline 1.13±0.09, after t-LTD 1.13±0.02 mV, n=5 (*Figure 1i and j*); MPP-GC - baseline 1.03±0.07, after t-LTD 1.00±0.09 mV, n=6 (*Figure 1s, t*). These results again suggest a presynaptic locus for t-LTD expression. Hence, together these results are consistent with a decrease in the probability of neurotransmitter release and they are indicative of a presynaptic locus for these two forms of t-LTD at both types of synapse.

## Presynaptic t-LTD requires NMDARs at MPP-GC but not at LPP-GC synapses

Different forms of t-LTD have been shown to require NMDARs at cortical synapses and in the hippocampus (*Bender et al., 2006*; *Nevian and Sakmann, 2006*; *Rodríguez-Moreno and Paulsen, 2008*; *Rodríguez-Moreno et al., 2011*; *Rodríguez-Moreno et al., 2013*; *Andrade-Talavera et al., 2016*). To determine whether the two forms of t-LTD studied here require NMDARs, we treated slices with the NMDAR antagonists D-AP5 (50 µM) or MK-801 (500 µM), completely blocking NMDARs and

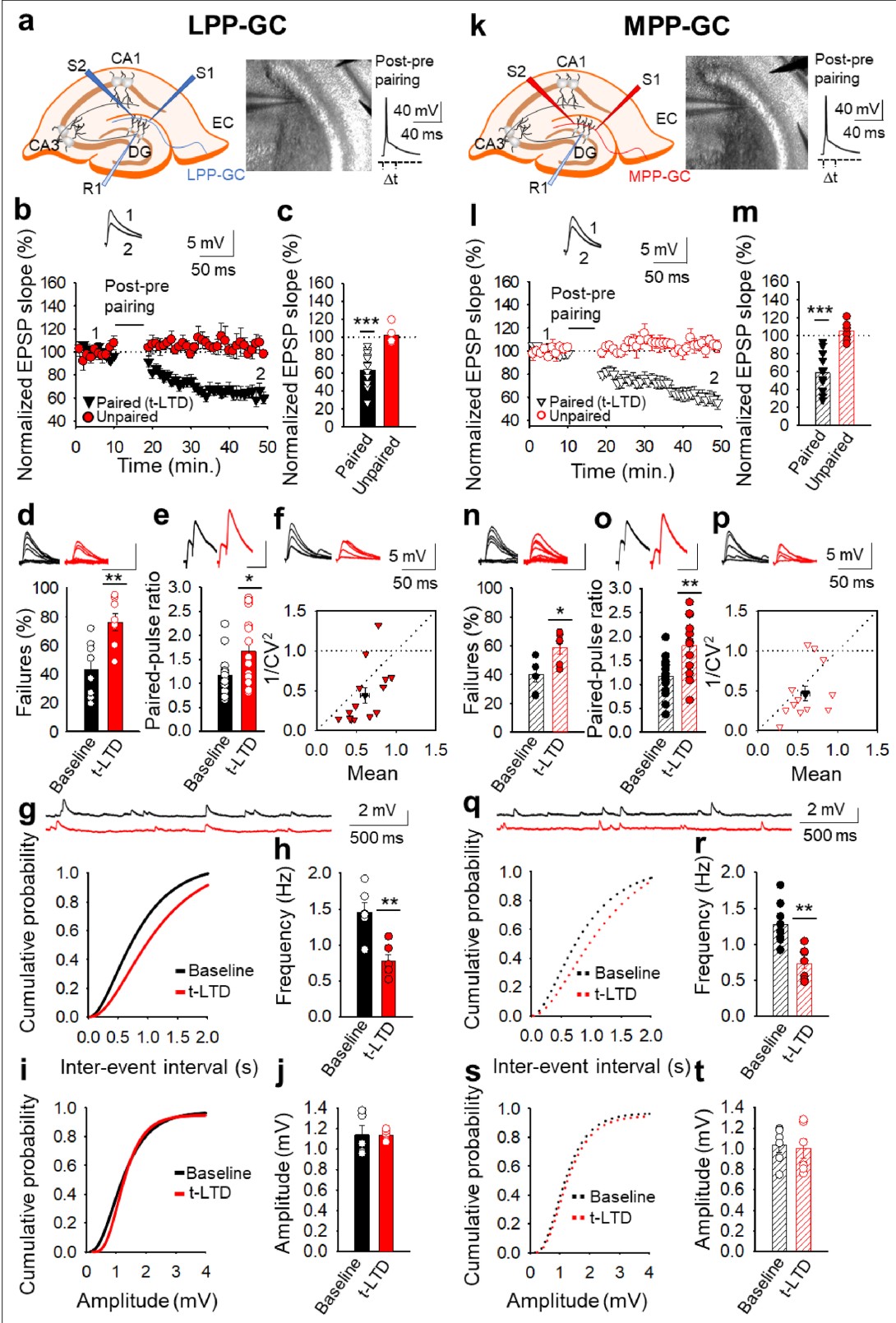

**Figure 1.** Input-specific presynaptic spike timing-dependent LTD at lateral and medial perforant pathway-dentate gyrus granule cell synapses. (**a, k**) Schemes and pictures show the general set-up for the electrophysiological recordings and pairing protocol (Δt = –18ms, time between the peak of the postsynaptic spike and the onset of the EPSP) at both LPP- and MPP-GC synapses: CA, *cornus ammonis*; DG, dentate gyrus; EC, entorhinal cortex; S1 and S2, stimulating electrodes; R, recording electrode. A post-before-pre pairing protocol induced t-LTD at both LPP-GC (**b**) and MPP-GC

*Figure 1 continued on next page*

*Figure 1 continued*

(**l**) synapses. The EPSP slopes monitored in the paired (LPP-GC synapses black triangles, n=14; MPP-GC synapses open black triangles, n=14) and unpaired (LPP-GC synapses red symbols; MPP-GC synapses open red symbols) pathways are shown. The traces show the EPSP before (1) and 30 min after (2) pairing. Depression was only observed in the paired pathways. (**c, m**) Summary of the results. *At the lateral (**d-j**) and medial (**n–t**) perforant pathway synapses onto dentate gyrus granule cells t-LTD is expressed presynaptically.* The number of failures increases after t-LTD induction at both LPP- (n=8, (**d**) and MPP-GC (n=6, (**n**) synapses. Traces show the EPSPs before (black) and 30 min after pairing (red). PPR increases after t-LTD pairing protocol at LPP-GC (**e**) and MPP-GC (**o**) synapses. The traces show the EPSP before (black) and 30 min after t-LTD induction (red). (**f, p**) Normalized plots of $CV^{-2}$ versus mean EPSP slopes suggest a presynaptic change in the release parameters at both LPP- (n=14) and MPP-GC (n=13) synapses. The traces show the EPSPs before (black) and 30 min after t-LTD induction (red). Miniature EPSPs (mEPSPs) were monitored at baseline and after t-LTD induction in the presence of TTX (300 nM) at LPP- (**g–j**), n=6) and MPP-GC synapses (**q–t**), n=6). Cumulative graphs and histograms show that after t-LTD induction, the frequency of mEPSPs decreases at both types of synapses, whereas the amplitude of the mEPSPs remains constant. Scale bar for the % failures - 5 mV, 50ms; scale bar for PPR - 5 mV, 80ms. Error bars indicate the S.E.M. * p<0.05, **p<0.01, *** p<0.001, two-sided Student's t-test.

The online version of this article includes the following source data and figure supplement(s) for figure 1:

**Source data 1.** Individual values included in the histograms.

**Figure supplement 1.** GABA receptors are not required for t-LTD at LPP- or MPP-GC synapses.

**Figure supplement 1—source data 1.** Individual values included in the histograms of *Figure 1—figure supplement 1*.

NMDAR-mediated currents (D-AP5: from 57±8 to 6±5 pA, n=5; MK-801: from 63±9 to 7±5 pA, n=5). In these experimental conditions, t-LTD at LPP-GC synapses was not affected by these antagonists (D-AP5 66 ± 4%, n=8; MK-801 62 ± 9%, n=6; interleaved controls 60 ± 6%, n=8: *Figure 2a and b*), whereas t-LTD at MPP-GC synapses was blocked in slices treated with either D-AP5 (90 ± 7 %, n=10) or MK-801 (107 ± 10 %, n=6) relative to the interleaved controls (60 ± 7 %, n=8: *Figure 2e and f*). When the experiment was repeated by loading the postsynaptic cell with MK-801 (500 μM-1 mM), t-LTD at MPP-GC synapses was not affected (54 ± 7 %, n=9) relative to the interleaved controls (60 ± 7 %, n=8: *Figure 2e and f*). MK-801 (500 μM-1 mM) loaded into the postsynaptic neuron blocked NMDAR-mediated currents recorded from the postsynaptic cell (from 58±5 to 9±5 pA, n=5). These results indicate that while t-LTD at MPP-GC synapses requires NMDARs, t-LTD at LPP-GC synapses does not. Moreover, the NMDARs required for t-LTD at MPP-GC synapses are ionotropic and non-postsynaptic, consistent with data from neocortical synapses where presynaptic NMDARs are required for t-LTD (*Sjöström et al., 2003*; *Bender et al., 2006*; *Nevian and Sakmann, 2006*; *Rodríguez-Moreno and Paulsen, 2008*).

Presynaptic NMDARs have been found at MPP-GC synapses where they modulate glutamate release. These receptors seem to contain GluN2B and GluN3A subunits (*Jourdain et al., 2007*; *Savtchouk et al., 2019*). To determine the subunit composition of the NMDARs involved in t-LTD at MPP-GC synapses, we first repeated the experiments in slices treated with $Zn^{2+}$ (300 nM), an antagonist of NMDARs containing GluN2A subunits (*Bidoret et al., 2009*; *Andrade-Talavera et al., 2016*; *Prius-Mengual et al., 2019*) that prevented t-LTD (95 ± 5 %, n=7: *Figure 2g and h*). We also repeated the experiments on slices in the presence of Ro25-6981 (0.5 μM), an antagonist of NMDARs containing GluN2B subunits (*Hu et al., 2009*; *Mareš et al., 2021*). This antagonist did not affect t-LTD (55 ± 11 %, n=6: *Figure 2g and h*) and neither did an additional antagonist of NMDARs containing GluN2B subunits, ifenprodil (3 μM, 62 ± 11%, n=6: *Figure 2g and h*). Finally, t-LTD was not affected in the presence of PPDA 10 μM, 61 ± 7%, n=6: *Figure 2g and h*, relative to the pooled interleaved controls (63 ± 6 %, n=9), an antagonist of NMDARs that contain GluN2C/2D subunits (*Chen et al., 2021*; *Wang et al., 2020*). These results indicate that the presynaptic NMDARs involved in t-LTD induction at MPP-GC synapses probably contain GluN1/GluN2A subunits.

## Presynaptic t-LTD at lateral and medial perforant pathway-dentate gyrus granule cell synapses requires mGluRs

As mGluRs have been implicated in plasticity and t-LTD in different brain regions, and at distinct synapses (*Gladding et al., 2009*), and they may modulate synaptic transmission at PP-DG synapses (*Dietrich et al., 1997*), we tested whether the presynaptic forms of t-LTD at LPP- and MPP-GC synapses require mGluRs. Exposure to the broad-spectrum mGluR antagonist LY341495 (100 μM) completely blocked t-LTD at both LPP-GC (100 ± 10 %, n=8: *Figure 2c and d*) and MPP-GC synapses (99 ± 9 %, n=10: *Figure 2i and j*). Indeed, the mGluR5 antagonist MPEP (20 μM) prevented t-LTD at LPP-GC (103 ± 10 %, n=8: *Figure 2c and d*) but not at MPP-GC synapses (61 ± 7 %, n=6: *Figure 2i and j*), whereas

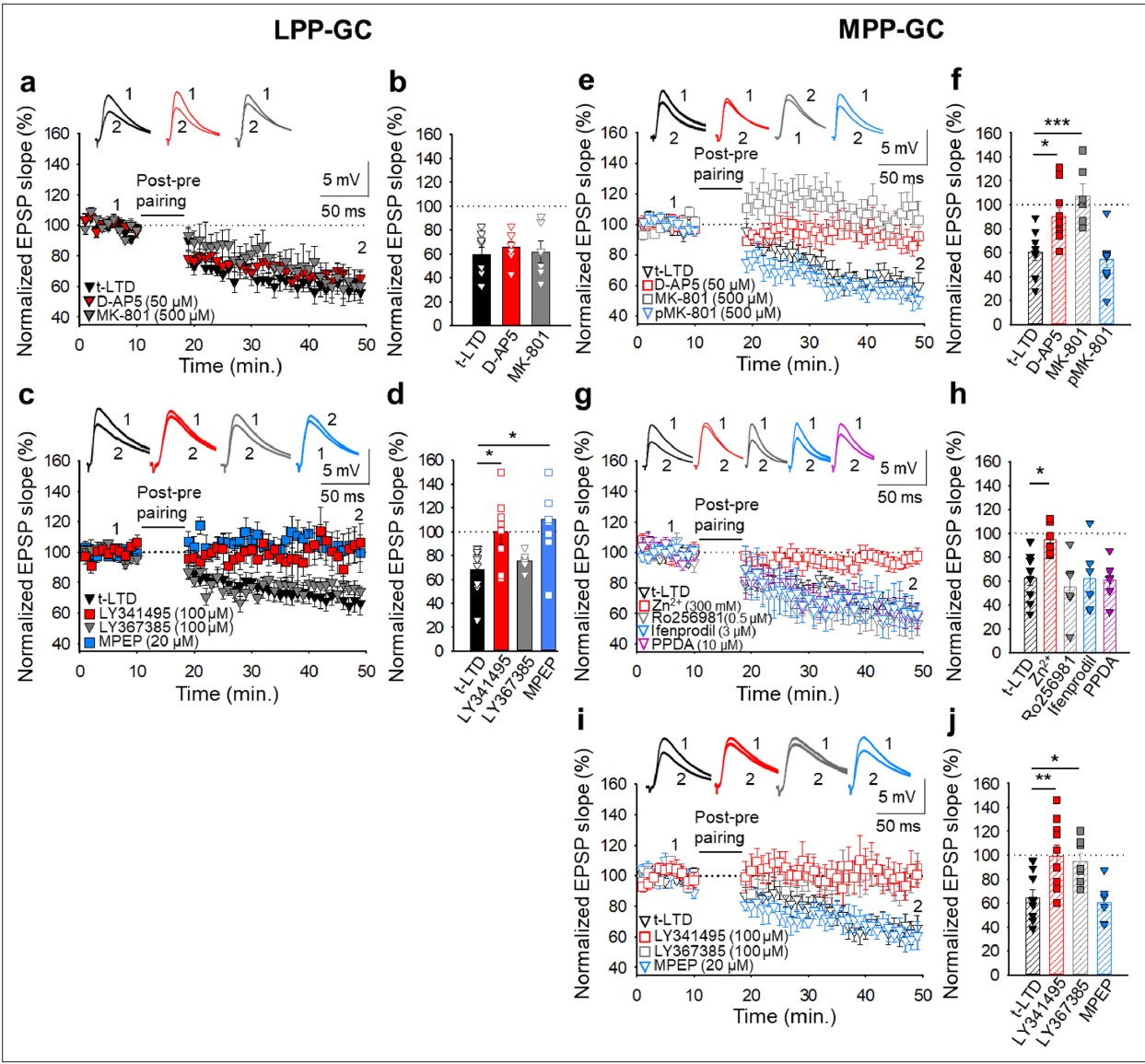

**Figure 2.** The t-LTD at lateral and medial perforant pathway-dentate gyrus granule cell synapses has distinct requirements for glutamate receptors. *The t-LTD at LPP-GC synapses does not require NMDAR but requires mGluR5.*(**a**) The addition of D-AP5 (50 µM) or MK-801 (500 µM) to the superfusion fluid does not prevent the induction of t-LTD at LPP-GC synapses. The EPSP slopes shown are from D-AP5-treated (red triangles, n=8), MK-801-treated (grey triangles, n=6) and untreated cells (black triangles, n=8). The traces show EPSPs before (1) and 30 min after (2) pairing. (**b**) Summary of the results. (**c**) The EPSP slopes are shown from control slices (black triangles, n=9), and slices treated with the mGluR antagonist LY341495 (100 µM, red squares, n=8), the mGluR1 antagonist LY367385 (100 µM, grey triangles, n=6) or the mGluR5 antagonist MPEP (20 µM, blue squares, n=8). The traces show the EPSPs before (1) and 30 min after (2) pairing. (**d**) Summary of the results. (**e**) *The t-LTD at MPP-GC synapses requires NMDARs containing GluN2A subunits and mGluR1.* The addition of D-AP5 (50 µM) or MK-801 (500 µM) to the superfusion fluid prevented t-LTD induction at MPP-GC synapses, whereas postsynaptic loading of MK-801 (500 µM) did not block t-LTD induction. The EPSP slopes are shown from D-AP5 (open red squares, n=10) or MK-801 treated cells (bath, open grey squares, n=6; loaded into postsynaptic cell, open blue triangles, n=9), and untreated cells (open black triangles, n=8). The traces show the EPSPs before (1) and 30 min after (2) pairing. (**f**) Summary of the results. (**g**) *The NMDARs involved in t-LTD at MPP-GC synapses contain GluN2A subunits.* The t-LTD at MPP-GC synapses was completely blocked in slices exposed to $Zn^{2+}$ (300 nM), while it remained unaffected in slices treated with Ro 25–6981 (0.5 µM), ifenprodil (3 µM) or PPDA (10 µM). The EPSP slopes shown are from control slices (open black triangles, n=9) and slices treated with $Zn^{2+}$ (open red squares, n=7), Ro-25–6981 (open grey triangles, n=6), ifenprodil (open blue triangles, n=6) or PPDA (open pink triangles, n=6). The traces show the EPSPs before (1) and 30 min after (2) pairing. (**h**) Summary of the results. (**i**) *The t-LTD at MPP-GC synapses requires mGluR1.* The EPSP slopes shown are from control slices (open black triangles, n=9), or slices treated with LY341495 (100 µM, open red squares, n=10), LY367385 (100 µM, open grey squares, n=7) or MPEP (20 µM, open blue triangles, n=6). The traces show the EPSPs before (1) and 30 min after (2) pairing. (**j**) Summary of the results. The error bars indicate the S.E.M. *p<0.05, ** p<0.01, ***p<0.001, One-way ANOVA + Holm–Sidak test.

The online version of this article includes the following source data for figure 2:

**Source data 1.** Individual values included in the histograms of *Figure 2*.

the specific mGluR1 antagonist LY367385 (100 µM) did not affect t-LTD at LPP-GC synapses (76 ± 3 %, n=6: *Figure 2c and d*) but it did block t-LTD at MPP-GC synapses (95 ± 7 %, n=7: *Figure 2i and j*) relative to their respective interleaved controls (LPP-GC synapses 68 ± 7%, n=9; MPP-GC synapses 65 ± 7%, n=9: *Figure 2c, d, i and j*). Hence, both LPP-GC and MPP-GC synapses require mGluR activity to induce these presynaptic forms of t-LTD, but whereas t-LTD at LPP-GC synapses requires mGluR5, mGluR1 receptors are required at MPP-GC synapses.

## Postsynaptic Ca²⁺ is required for t-LTD at lateral and medial perforant pathway-dentate gyrus granule cell synapses

Both t-LTP and t-LTD require postsynaptic $Ca^{2+}$ at neocortical and hippocampal synapses (*Bender et al., 2006*; *Nevian and Sakmann, 2006*; *Rodríguez-Moreno et al., 2013*; *Andrade-Talavera et al., 2016*; *Inglebert and Debanne, 2021*; *Martínez-Gallego et al., 2022a*), which led us to investigate the postsynaptic $Ca^{2+}$ requirements of t-LTD at PP-DG synapses. As L-type $Ca^{2+}$ channels have previously been implicated in plasticity (*Nevian and Sakmann, 2006*; *Andrade-Talavera et al., 2016*; *Wiera et al., 2017*; *Falcón-Moya et al., 2020*; *Martínez-Gallego et al., 2022a*), we assessed whether they are involved in the t-LTD studied here by performing the pairing protocol after bath application of the L-type $Ca^{2+}$ channel blocker, nimodipine (10 µM). This channel blocker did not impede t-LTD induction at LPP-GC (79 ± 4 %, n=6: *Figure 3a and b*) or MPP-GC synapses (67 ± 6 %, n=6: *Figure 3e and f*), indicating that L-type $Ca^{2+}$ channels are not required for these forms of t-LTD. The $Ca^{2+}$ involved in synaptic plasticity may also be released from intracellular stores (*Bender et al., 2006*; *Nevian and Sakmann, 2006*; *Andrade-Talavera et al., 2016*; *Martínez-Gallego et al., 2022a*) and indeed, t-LTD was prevented at both LPP-GC and MPP-GC synapses when 10 µM thapsigargin (a drug that avoids refilling of intracellular $Ca^{2+}$ stores after $Ca^{2+}$ depletion, *Singh et al., 2021*) was present in the superfusion fluid (LPP-GC synapses 108 ± 16%, n=6; MPP-GC synapses 108 ± 6 %, n=6),. A similar effect was witnessed when thapsigargin (1–10 µM) was loaded into the postsynaptic neuron (LPP-GC synapses 107 ± 7%, n=15; MPP-GC synapses 98 ± 4%, n=13) relative to untreated interleaved controls (LPP-GC synapses 65 ± 5%, n=11; MPP-GC synapses 63±6, n=11: *Figure 3a, b, e and f*). The presence of heparin (5 mg/ml), a blocker of $IP_3R$-mediated $Ca^{2+}$ release (*Ghosh et al., 1988*), in the recording pipette did not prevent t-LTD induction at LPP- (77 ± 8%, n=6; versus interleaved control slices, 65 ± 5%, n=11; *Figure 3a and b*) or MPP-GC synapses (81 ± 10%, n=6; versus interleaved control slices, 63 ± 6%, n=11; *Figure 3e and f*), suggesting that postsynaptic $IP_3R$-mediated $Ca^{2+}$ release is not required for t-LTD. In contrast, inclusion in the patch pipette of ruthenium red (a blocker of ryanodine receptors), prevented t-LTD at both, LPP- (98 ± 12%, n=7; versus interleaved control slices, 65 ± 5%, n=11, *Figure 3a and b*) and MPP-GC synapses (98 ± 12%, n=6; versus interleaved control slices, 63 ± 6%, n=11, *Figure 3e and f*), suggesting that $Ca^{2+}$ release from ryanodine-sensitive $Ca^{2+}$ stores is required for this form of t-LTD. Hence, $Ca^{2+}$ release from the intracellular stores appears to be necessary for t-LTD at both LPP and MPP-GC synapses.

## Postsynaptic calcineurin is required for t-LTD induction at LPP-GC but not MPP-GC synapses

To gain better insight into the mechanisms involved in t-LTD, we hypothesized that a $Ca^{2+}$-dependent enzyme might be involved. Since the $Ca^{2+}$-dependent protein phosphatase calcineurin has been implicated in synaptic plasticity in the hippocampus (*Mulkey et al., 1994*) and neocortex (*Torii et al., 1995*; *Rodríguez-Moreno et al., 2013*; *Andrade-Talavera et al., 2016*), we tested the effect of inhibiting calcineurin on t-LTD. Accordingly, t-LTD induction was prevented at LPP-GC (110 ± 7 %, n=8) but not MPP-GC synapses (72 ± 7 %, n=6: *Figure 3—figure supplement 1*) when the calcineurin blocker FK506 (10 µM) was added to the superfusion fluid, relative to the untreated interleaved controls (LPP-GC: 68 ± 6%, n=9; MPP-GC: 71 ± 7%, n=6: *Figure 3—figure supplement 1*). Subsequently, we loaded the postsynaptic neuron at LPP-GC synapses with FK506 (1–10 µM) and as t-LTD was no longer detected at these synapses (109 ± 8%, n=13: *Figure 3—figure supplement 1*), postsynaptic calcineurin appears to be involved in t-LTD induction at LPP-GC but not MPP-GC synapses.

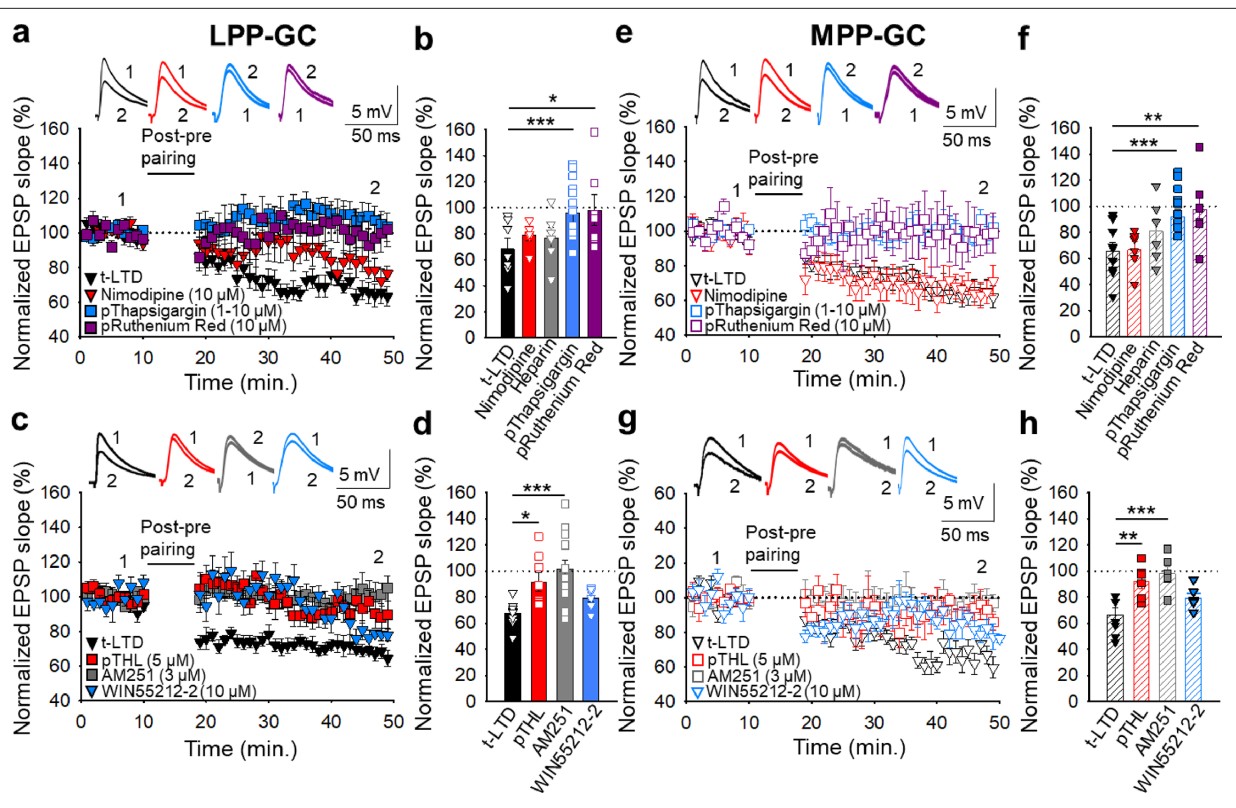

**Figure 3.** The t-LTD at lateral and medial perforant pathway-dentate gyrus granule cell synapses requires postsynaptic calcium and eCB signalling. (**a**) The induction of t-LTD at LPP-GC synapses in slices was unaffected by exposure to nimodipine (10 µM) or to heparin (5 mg/ml), but it was prevented by treatment with thapsigargin (1–10 µM) or ruthenium red (10 µM), loaded into the postsynaptic cell. The EPSP slopes shown are from control slices (black triangles, n=11) and slices treated with nimodipine (red triangles, n=6), thapsigargin (blue squares, n=15) and ruthenium red (purple squares, n=7). The traces show EPSPs before (1) and 30 min after (2) pairing. (**b**) Summary of the results. (**c**) The t-LTD at LPP-GC synapses requires endocannabinoids and CB1 receptors, as it was blocked when THL was loaded into the postsynaptic neuron (pTHL, 5 µM) or in slices treated with AM251 (3 µM). In addition, LTD was induced by direct activation of $CB_1R$ by treatment of the slices with WIN 55 512–2 (10 µM). The EPSP slopes shown are from control slices (black triangles, n=9), in slices with THL loaded into the postsynaptic neuron (red squares, n=8), treated with AM251 (grey squares, n=14) or treated with WIN 55 512–2 (blue triangles, n=6). The traces show the EPSPs before (1) and 30 min after (2) pairing. (**d**) Summary of the results. (**e**) The induction of t-LTD at MPP-GC synapses was unaffected in nimodipine-treated slices (10 µM) or heparin (5 mg/ml), but prevented in slices treated with thapsigargin (10 µM) or ruthenium red (10 µM), loaded into the postsynaptic cell. The EPSP slopes shown are from control slices (open black triangles, n=9) and slices treated with nimodipine (open red triangles, n=6), thapsigargin (open blue squares, n=13) and ruthenium red (open purple squares, n=6). The traces show EPSPs before (1) and 30 min after (2) pairing. (**f**) Summary of the results. (**g**) The t-LTD at MPP-GC synapses requires endocannabinoids and $CB_1$ receptors, and it was blocked when THL was loaded into the postsynaptic neuron (pTHL, 5 µM) and in slices treated with AM251 (3 µM). In addition, LTD was induced by direct activation of $CB_1R$ by treatment of the slices with WIN 55 512–2 (10 µM). The EPSP slopes shown are from control slices (open black triangles, n=8) and slices with THL loaded into the postsynaptic neuron (open red squares, n=6) or treated with AM251 (open grey squares, n=6) or treated with WIN 55 512–2 (open blue triangles, n=6). The traces show EPSPs before (1) and 30 min after (2) pairing. (**h**) Summary of the results. The error bars indicate S.E.M. *p<0.05, ** p<0.01, *** p<0.001, One-way ANOVA + Holm–Sidak test.

The online version of this article includes the following source data and figure supplement(s) for figure 3:

**Source data 1.** Individual values included in the histograms of *Figure 3*.

**Figure supplement 1.** Calcineurin is required for t-LTD induction at the LPP, but not at MPP-GC synapses.

**Figure supplement 1—source data 1.** Individual values included in the histograms of *Figure 3—figure supplement 1*.

## Endocannabinoid signalling at lateral and medial perforant pathway-dentate gyrus granule cell synapses is required for t-LTD

Postsynaptic $Ca^{2+}$ dynamics may affect the synthesis of eCBs like anandamide and 2-arachidonoylglycerol (2-AG), ligands of $CB_1Rs$ (*Laguerre et al., 2021*) necessary for t-LTD at neocortical, striatal and hippocampal synapses in rodents (*Bender et al., 2006*; *Andrade-Talavera et al., 2016*; *Cui et al., 2016*; *Hogrefe et al., 2022*). These receptors may be expressed in neurons and/or astrocytes, where they can mediate the release of gliotransmitters like glutamate, ATP/adenosine and D-serine through

exocytotic mechanisms (*Covelo and Araque, 2018*). The induction of t-LTD at hippocampal and neocortex synapses requires the eCB system and astrocyte signalling, which may in turn drive gliotransmitter release (*Min and Nevian, 2012*; *Andrade-Talavera et al., 2016*; *Martínez-Gallego et al., 2022a*). To investigate the role of eCB signalling in these forms of t-LTD, we loaded the postsynaptic neurons with tetrahydrolipstatin (THL, 5 µM) to inhibit diacylglycerol lipase activity and prevent eCB synthesis. THL prevented t-LTD induction at both LPP-GC (92 ± 7 %, n=8 versus interleaved controls 67±3% n=8: *Figure 3c and d*) and MPP-GC synapses (92 ± 5 %, n=6 versus interleaved controls 66 ± 5%, n=8: *Figure 3g and h*). Moreover, the presence of the $CB_1R$ antagonist AM251 (3 µM) also blocked t-LTD at both LPP-GC (102 ± 7 %, n=14: *Figure 3c and d*) and MPP-GC synapses (98 ± 5 %, n=6: *Figure 3g and h*). In addition, the activation of $CB_1R$ by puffs of the agonist WIN 55, 212–2 to the astrocyte, directly induced LTD at both LPP- (83 ± 9 %, n=6 versus interleaved controls 68±3% n=9: *Figure 3c and d*) and MPP-GC synapses (79 ± 3 %, n=6 versus interleaved controls 67±5% n=8: *Figure 3g and h*). Thus, eCB synthesis by postsynaptic cells and $CB_1R$ activation is required for presynaptic t-LTD at LPP- and MPP-DG GC synapses.

## Astrocyte signalling and glutamate release at lateral and medial perforant pathway-dentate gyrus granule cell synapses is required for t-LTD

Astrocytes are involved in plasticity and synaptic transmission at hippocampal and cortical synapses (*Min and Nevian, 2012*; *Araque et al., 2014*; *Andrade-Talavera et al., 2016*; *Covelo and Araque, 2018*; *Pérez-Rodríguez et al., 2019*; *Falcón-Moya et al., 2020*; *Durkee et al., 2021*; *Martínez-Gallego et al., 2022a*), and at the synapses studied here (*Jourdain et al., 2007*; *Savtchouk et al., 2019*; *Pérez-Otaño and Rodríguez-Moreno, 2019*). To investigate the possible involvement of astrocytes (*Figure 4—figure supplement 1*) in the induction of t-LTD at the EC-DG synapses studied here, different approaches were used. First, individual astrocytes were loaded with the $Ca^{2+}$ chelator BAPTA (20 mM) though the patch pipette to inhibit vesicular and $Ca^{2+}$-dependent gliotransmitter release (*Parpura and Zorec, 2010*), both at LPP-GC and MPP-GC synapses, and recorded in neurons situated at 50–100 µm from the loaded astrocytes (*Figure 4a and f*). In these experimental conditions, t-LTD was impaired at both LPP-GC (103 ± 9 %, n=6: *Figure 4b and c*) and MPP-GC synapses (94 ± 6 %, n=6: *Figure 4g and h*). Second, we assessed t-LTD in transgenic mice expressing a selective dnSNARE domain (dnSNARE) in astrocytes, preventing functional vesicular gliotransmitter release from these cells (*Pascual et al., 2005*; *Sultan et al., 2015*; *Sardinha et al., 2017*). In contrast to the typical t-LTD observed in WT mice (LPP-GC synapses 71 ± 5%, n=7; MPP-GC synapses 60 ± 5%, n=6: *Figure 4b, c, g and h*), t-LTD could not be induced in these dnSNARE mice (LPP-GC synapses 102 ± 9%, n=9; MPP-GC synapses 112 ± 6%, n=7: *Figure 4c, d, g and h*). In addition, we loaded astrocytes with the light chain of the tetanus toxin (TeTx$_{LC}$, 1 µM) which is known to block exocytosis by cleaving the vesicle-associated membrane protein, an important part of the SNARE complex (*Schiavo et al., 1992*; *Min and Nevian, 2012*). In these experimental conditions, t-LTD was impaired at both LPP-GC (106 ± 9 %, n=6: *Figure 4b and c*) and MPP-GC synapses (108 ± 10 %, n=6: *Figure 4g and h*). These results clearly indicate that astrocytes and $Ca^{2+}$-dependent vesicular release are required to induce t-LTD at both LPP-GC and MPP-GC synapses.

At these synapses, it is thought that glutamate is released from astrocytes to modulate synaptic transmission and participate in plasticity (*Jourdain et al., 2007*; *Savtchouk et al., 2019*), as indicated for cortical synapses (*Min and Nevian, 2012*; *Martínez-Gallego et al., 2022a*). To determine whether glutamate, probably released by astrocytes, is required for t-LTD at LPP- and MPP-GC synapses, we repeated the experiments in slices from dnSNARE mutant mice but applying glutamate puffs (100 µM) in the proximity of the astrocytes and neurons under study. In these experimental conditions, t-LTD, which was not evident in untreated dnSNARE mice (LPP-GC synapses 95 ± 5%, n=7; MPP-GC synapses 109 ± 6%, n=6) was fully recovered at both LPP-GC (72 ± 11 %, n=6: *Figure 4d and e*) and MPP-GC synapses (70 ± 8 %, n=6: *Figure 4i and j*) relative to the control interleaved slices from WT mice (LPP-GC synapses 62 ± 8%, n=7; MPP-GC synapses 64 ± 5%, n=7). In addition, to gain more insight into the fact that glutamate is released by astrocytes, we blocked glutamate release from astrocytes by loading the astrocytes with Evans blue (5 µM), known to interfere with glutamate uptake into vesicles as it inhibits the vesicular glutamate transporter (VGLUT) (*Harkany et al., 2004*; *Min and Nevian, 2012*). In these experimental conditions, t-LTD was impaired at both LPP-GC (106 ± 6 %, n=6:

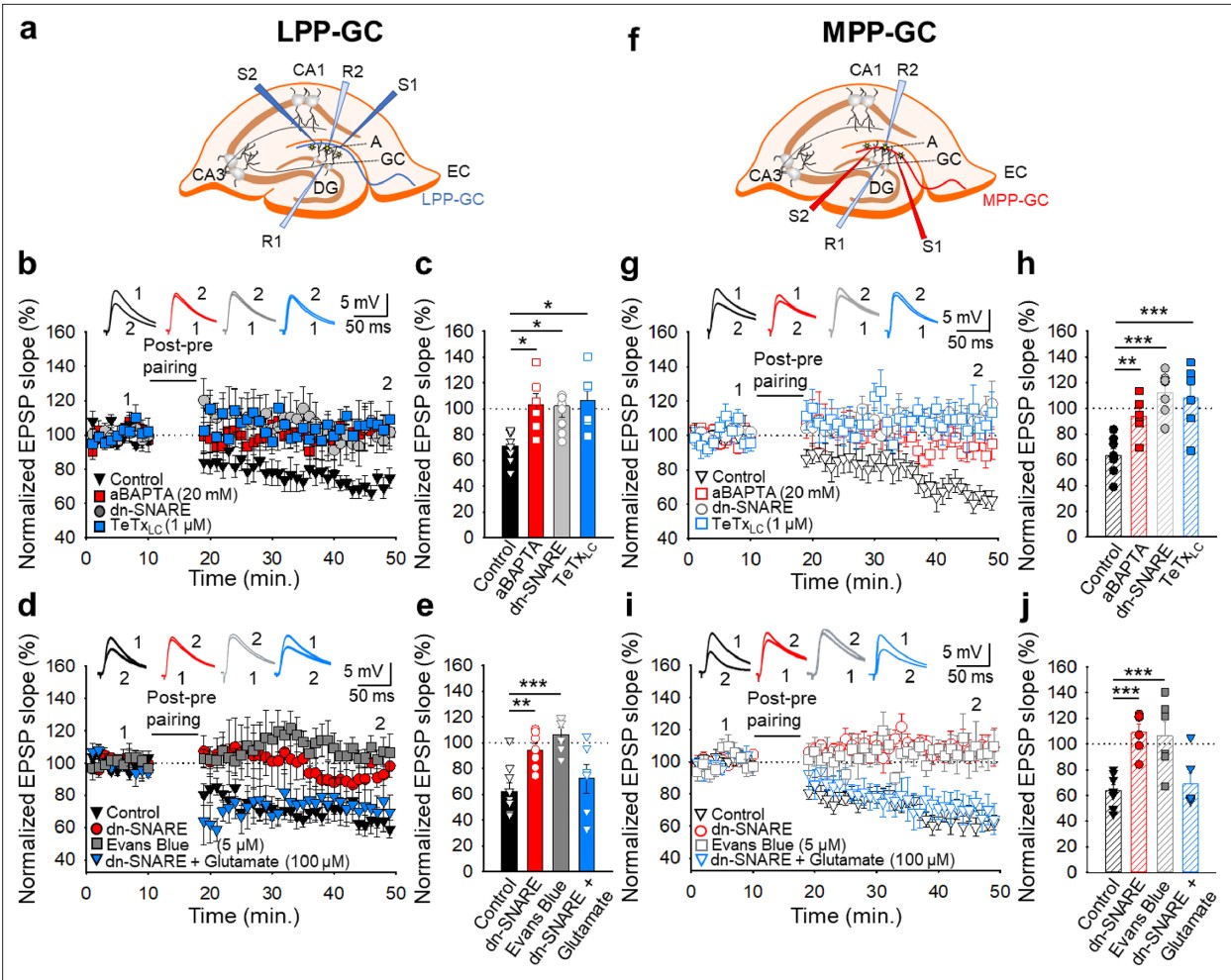

**Figure 4.** The t-LTD at lateral and medial perforant pathway-dentate gyrus granule cell synapses requires astrocytes and glutamate. (**a, f**) Schemes showing the general experimental set-up: A, astrocyte; CA, *cornus ammonis*; DG, dentate gyrus; EC, entorhinal cortex; GC, granule cell; S1 and S2, stimulating electrodes; R1 and R2, recording electrodes. (**b**) The t-LTD at LPP-GC synapses requires astrocytes as it was prevented by loading astrocytes with BAPTA (20 mM) and in slices from dnSNARE mice or when astrocytes are treated with the light chain of the tetanus toxin (TeTx$_{LC}$, 1 μM). The EPSP slopes shown are from control slices (black triangles, n=7), slices with BAPTA-loaded astrocytes (red squares, n=6), slices from dnSNARE mice (grey circles, n=9) and in slices treated with TeTx$_{LC}$ (blue squares, n=6). The traces show the EPSPs before (1) and 30 min after (2) pairing. (**c**) Summary of the results. (**d**) The t-LTD at LPP-GC synapses requires glutamate as it is absent in slices from dnSNARE mice but it is restored by glutamate puffs (100 μM). Note that t-LTD is also absent in slices treated with Evans blue (5 μM). The EPSP slopes shown are from control slices (black triangles, n=8), slices from dnSNARE mice (red circles, n=7), slices from dnSNARE mice administered glutamate puffs (blue triangles, n=6), and slices treated with Evans blue loaded into the astrocytes (grey squares, n=6). The traces show the EPSPs before (1) and 30 min after (2) pairing. (**e**) Summary of the results. (**g**) The t-LTD at MPP-GC synapses requires astrocytes as t-LTD induction at MPP-GC synapses was prevented by loading astrocytes with BAPTA (20 mM), in slices from dnSNARE mice or when astrocytes are treated with the light chain of the tetanus toxin (TeTx$_{LC}$, 1 μM). The EPSP slopes shown are from control slices (open black triangles, n=6), slices with BAPTA-loaded astrocytes (open red squares, n=6), slices from dnSNARE mice (open grey circles, n=7) and in slices treated with TeTx$_{LC}$ (open blue squares, n=6). The traces show the EPSPs before (1) and 30 min after (2) pairing. (**h**) Summary of the results. (**i**) The t-LTD at MPP-GC synapses requires glutamate as it is absent in slices from dnSNARE mice but is restored by glutamate puffs (100 μM). Note that t-LTD is also absent in slices treated with Evans blue (5 μM). The EPSP slopes shown are from control slices (open black triangles, n=8), slices from dnSNARE mice (open red circles, n=6) slices from dnSNARE mice administered to glutamate puffs (open blue triangles, n=6) and slices treated with Evans blue loaded into the astrocytes (open grey squares, n=6). The traces show the EPSPs before (1) and 30 min after (2) pairing. (**j**) Summary of the results. The error bars indicate the S.E.M. *p<0.05, **p<0.01, ***p<0.001, One-way ANOVA + Holm–Sidak test.

The online version of this article includes the following source data and figure supplement(s) for figure 4:

**Source data 1.** Individual values included in the histograms of *Figure 4*.

**Figure supplement 1.** Identification of astrocytes.

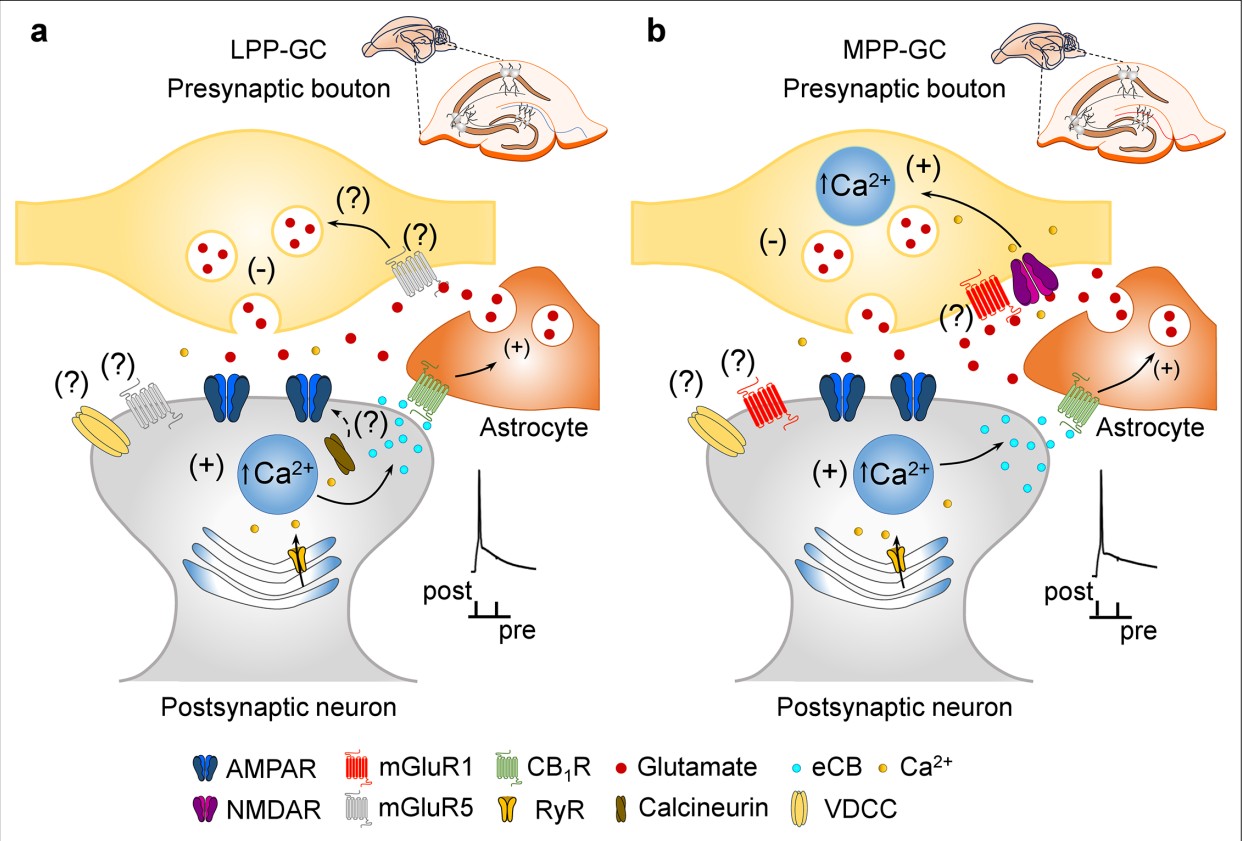

**Figure 5.** Model of presynaptic t-LTD at LPP- and MPP-GC synapses. t-LTD is induced by a post-before-pre, single-spike pairing protocol at P13-21. Postsynaptic action potentials depolarize the postsynaptic membrane that probably permeates $Ca^{2+}$ into the postsynaptic neuron, causing calcineurin activation (at LPP-GC synapses) and $Ca^{2+}$ release from internal stores in a $Ca^{2+}$-dependent $Ca^{2+}$ release manner at both, LPP- and MPP-GC synapses, driving eCBs synthesis and release. The eCB signal leads to the activation of $CB_1$ receptors, possibly situated in surrounding astrocytes, facilitating glutamate release from astrocytes, which, together with glutamate released from presynaptic neurons, that at LPP-GC synapses (**a**) possibly activates mGluR5 and presynaptic NMDA receptors at MPP-GC synapses (**b**) to produce a decrease in glutamate release probability in both cases and t-LTD.

*Figure 4d and e*) and MPP-GC synapses (106 ± 11 %, n=6: *Figure 4i and j*). These results indicate that glutamate released involving SNARE-dependent exocytosis by astrocytes is necessary for the induction of t-LTD at LPP- and MPP-GC synapses.

## Discussion

In this study, two new forms of t-LTD were seen to be elicited at LPP and MPP synapses onto DG GCs by pairing presynaptic activity with single postsynaptic APs at low frequency (0.2 Hz). Using different approaches, we found that these two forms of t-LTD are expressed presynaptically through a decrease in the glutamate release probability. Interestingly, t-LTD induction at MPP-GC synapses but not at LPP-GC synapses depends on presynaptic NMDARs containing GluN2A subunits. In addition, mGluRs are required for t-LTD at both types of PP-DG synapses, although LPP-GC synapses require mGluR5 activation and MPP-GC synapses require mGluR1 activity. These two forms of presynaptic t-LTD require postsynaptic $Ca^{2+}$ released from postsynaptic intracellular stores and whereas t-LTD at LPP-GC synapses, requires calcineurin phosphatase activity; this is not the case at MPP-GC synapses. These two forms of t-LTD require eCB synthesis and $CB_1R$ activation, and both require astrocyte signalling to promote glutamate release (*Figure 5*).

## Pairing presynaptic activity with single postsynaptic action potentials at low frequency can induce presynaptic t-LTD at mouse lateral and medial perforant pathway-dentate gyrus granule cell synapses

While different time intervals between the postsynaptic spike and presynaptic EPSP of a pairing protocol may induce t-LTD (*Feldman, 2012*), it is clear here that pairing EPSPs of 3–5 mV size 100 times with single postsynaptic spikes 18ms later at 0.2 Hz induces robust LTD at both LPP- and MPP-GC synapses. Interestingly, a similar protocol induces t-LTD at hippocampal CA3-CA1 synapses (*Andrade-Talavera et al., 2016*; *Pérez-Rodríguez et al., 2019*; *Falcón-Moya et al., 2020*) and at L4-L2/3 synapses of the somatosensory (*Rodríguez-Moreno and Paulsen, 2008*; *Banerjee et al., 2009*; *Banerjee et al., 2014*; *Martínez-Gallego et al., 2022a*), visual (*Larsen et al., 2014*) and cingulate cortices (*Hogrefe et al., 2022*). As such, the results presented here extend the existence of t-LTD to LPP and MPP synapses onto hippocampal DG GCs. Although there are several uncertainties associated with quantal analysis in the CNS (see *Brock et al., 2020*), we used different approaches here to determine the locus of expression of these two forms of t-LTD (failure rate, PPRs, CV, and mEPSP frequency and amplitude analysis), all of which were consistent with presynaptic changes.

## Presynaptic t-LTD requires NMDARs at MPP-GC but not at LPP-GC synapses

NMDARs are required for general synaptic plasticity and for STDP in different experimental models (cell cultures, rodent brain slices and human neurons). They are required for t-LTD and t-LTP at different synapses in the hippocampus, neocortex, cerebellum and striatum (*Bi and Poo, 1998*; *Bender et al., 2006*; *Nevian and Sakmann, 2006*; *Banerjee et al., 2009*; *Banerjee et al., 2014*; *Fino et al., 2010*; *Verhoog and Mansvelder, 2011*; *Kwag and Paulsen, 2012*; *Andrade-Talavera et al., 2016*; *Sgritta et al., 2017*; *Louth et al., 2021*; *Martínez-Gallego et al., 2022a*). NMDARs have been implicated in synaptic plasticity at MPP-GC but not LPP-GC synapses through classical protocols (*Dahl et al., 1990*; *Min et al., 1998*; *Beck et al., 2000*; *Hayashi et al., 2014*). Our results are consistent with the presence of NMDARs at MPP-GC synapses, in our case participating in LTD. Indeed, t-LTD is not present at MPP-GC synapses in the presence of D-AP5, and the presence of presynaptic NMDARs at MPP-GC synapses was demonstrated previously by immunogold labelling (*Jourdain et al., 2007*; *Savtchouk et al., 2019*). At these synapses, t-LTD was also blocked when MK801 was present in the bath but not when it was loaded into the postsynaptic neuron, indicating that the NMDARs necessary for t-LTD are not postsynaptic, as seen in L4-L2/3 synapses of the somatosensory (*Bender et al., 2006*; *Nevian and Sakmann, 2006*; *Rodríguez-Moreno and Paulsen, 2008*; *Rodríguez-Moreno et al., 2011*; *Rodríguez-Moreno et al., 2013*) and visual (*Larsen et al., 2014*) cortices. Since the NMDA channel blocker MK801 prevents t-LTD, the NMDARs required for t-LTD appear to be ionotropic, although an additional role for a postsynaptic metabotropic NMDARs cannot be ruled-out (*Nabavi et al., 2013*). Thus, the involvement of NMDARs is one important difference between LPP- and MPP-GC synapses that may account for at least some of the physiological differences described between these two pathways. By using antagonists with different NMDAR subunit specificities, we also found that the NMDARs required for t-LTD at MPP-GC synapses contain GluN2A but not GluN2B or GluN2C/D subunits. NMDARs that enhance glutamate release or that participate in LTP at PP-GC synapses are thought to contain GluN2B subunits in rats as their effect is blocked by ifenprodil (*Jourdain et al., 2007*; *Savtchouk et al., 2019*). However, ifenprodil or Ro25698 had no effect on t-LTD here, such that the preNMDARs containing GluN2B subunits that participate in LTP and enhance glutamate release may differ from those participating in t-LTD, and there may even be differences between the species (i.e.: rat and mouse).

## Presynaptic t-LTD at lateral and medial perforant pathway-dentate gyrus granule cell synapses requires mGluRs

While both forms of t-LTD described here appear to require group I mGluRs, they require different ones: mGluR5 at LPP-GC synapses and mGluR1 at MPP-GC contacts. Nevertheless, the exact location of these mGluRs is not clear at present, which may represent a further difference between LPP- and MPP-GC synapses, and with important behavioural and physiological consequences (*Martínez-Gallego et al., 2022b*).

## Postsynaptic Ca²⁺ is required for t-LTD at lateral and medial perforant pathway-dentate gyrus granule cell synapses

Both, t-LTD and t-LTP require postsynaptic $Ca^{2+}$ at neocortical (*Bender et al., 2006*; *Nevian and Sakmann, 2006*; *Rodríguez-Moreno et al., 2013*; *Martínez-Gallego et al., 2022a*) and hippocampal synapses (*Andrade-Talavera et al., 2016*; *Falcón-Moya et al., 2020*; *Mateos-Aparicio and Rodríguez-Moreno, 2020*; *Inglebert and Debanne, 2021*). Moreover, like other forms of t-LTD, that at LPP- and MPP-GC synapses requires a rise in $Ca^{2+}$ into the postsynaptic cell due to $Ca^{2+}$ release from internal stores, as also seen in other situations of hippocampal, neocortical and cerebellar t-LTD (*Bender et al., 2006*; *Nevian and Sakmann, 2006*; *Andrade-Talavera et al., 2016*; *Sgritta et al., 2017*; *Falcón-Moya et al., 2020*; *Martínez-Gallego et al., 2022a*). This is similar to the LTD induced by LFS in DG GCs (*Wang and Kelly, 1997*). Our data also implies an involvement of calcineurin in t-LTD induction at LPP-GC synapses but not at MPP-GC synapses, an additional striking difference between these two types of synapses. Protein phosphatases, including calcineurin, may be required for different forms of LTD, both in the hippocampus (*Mulkey et al., 1994*) and neocortex (*Torii et al., 1995*). Future experiments will determine the exact mechanism by which calcineurin mediates t-LTD at LPP but not at MPP-GC synapses. What is clear for the moment is that while the requirements for $Ca^{2+}$ seem to be similar at both LPP-GC and MPP-GC synapses, the $Ca^{2+}$-dependent protein phosphatase calcineurin is only involved in t-LTD at LPP-GC synapses. Significantly, the functional consequences of this difference remain to be determined.

## Endocannabinoid signalling at lateral and medial perforant pathway-dentate gyrus granule cell synapses is required for t-LTD

Our results indicate that eCBs and $CB_1Rs$ are involved in the induction of these two different forms of t-LTD, in agreement with previous field recordings showing that the eCB system is involved in synaptic plasticity at PP-DG synapses (*Wang et al., 2016*; *Peñasco et al., 2019*; *Fontaine et al., 2020*). Here, we did not elucidate the location of the $CB_1Rs$ mediating t-LTD, such that further work will be required to address this question. However, the demonstration that astrocytes are required for t-LTD and that WIN55, 212–2 puffs over astrocytes induce LTD are suggestive that astrocytic $CB_1Rs$ might be involved.

## Astrocyte signalling and glutamate release at lateral and medial perforant pathway-dentate gyrus granule cell synapses is required for t-LTD

Through four different approaches (introducing BAPTA, tetanus toxin light chain or Evans blue into astrocytes and using mutant dnSNARE mice), we demonstrated the involvement of astrocytes in presynaptic t-LTD, both at LPP- and MPP-GC synapses. Astrocytes participate in synaptic potentiation at PP-GC synapses (*Jourdain et al., 2007*), although little is known about the contribution of astrocytes to LTD. Thus, this is the first demonstration of the participation of astrocytes in LTD at these synapses. Interestingly, and as found in the somatosensory cortex (*Min and Nevian, 2012*), glutamate is required to mediate t-LTD induction at both types of PP-GC synapses, and in fact, t-LTD is impaired when astrocytes were loaded with Evans blue and recovered by the local application of glutamate at both LPP- and MPP-GC synapses in slices from dnSNARE mice that do not develop t-LTD otherwise. These findings agree with previous studies showing that astrocytes can release glutamate that reaches the NMDARs located in the presynaptic boutons at PP-GC synapses (*Bezzi et al., 2004*; *Jourdain et al., 2007*), influencing synaptic transmission at MPP-GC synapses (*Savtchouk et al., 2019*). It is interesting to note that the presynaptic targets of the glutamate, at least in part released by astrocytes, are not the same at LPP- and MPP-GC synapses, as they are mGluRs in the case of LPP-GC synapses, and both NMDARs and mGluRs in the case of MPP-GC synapses. While it is possible that glutamate from the presynaptic neuron also activates preNMDARs and mGluRs, the amount of glutamate released by presynaptic terminals may be insufficient to drive t-LTD, and hence, glutamate released from astrocytes may be required for t-LTD.

## What is the physiological role of these forms of plasticity?

For the moment, the exact role of STDP in the perforant cortex or in the hippocampus is not known, and more work is necessary to specifically determine functions for t-LTP and t-LTD. STDP seems a good candidate to mediate spatial learning in the hippocampus (*Bush et al., 2010*) and the possible role of t-LTP and t-LTD in forms of learning involving the hippocampus and the perforant cortex will be addressed in future studies. Our studies were performed in developing P13-P21 animals indicating its relevance during maturation. The functions of t-LTP and t-LTD during development are most probably related to the refinement of synaptic connections and remodelling of neuronal circuits (*Andrade-Talavera et al., 2023*). As a Hebbian learning rule, t-LTP should occur when the spike order is pre-before-post, strengthening those connections in which the presynaptic neuron takes part in firing the postsynaptic cell, as predicted by Hebb, whereas t-LTD occurs when the spike order is reversed, so that non-causal spiking weakens the connections involved, possibly as a first step in the elimination of those connections during development as has been suggested (see *Bush et al., 2010*). Indeed, further studies will be necessary to determine the true influence of STDP in the perforant cortex and in the hippocampus, and the specific developmental role of t-LTD and t-LTP in these circuits.

It is interesting to note that with these new results it is clear that both forms of t-LTD may require glutamate from astrocytes. Presynaptic plasticity may involve structural changes and may change the short-term properties of neurotransmitter release, participating in circuit computations, and changing the excitatory/inhibitory balance or sensory adaptations (*Monday et al., 2018*). Why some synapses, like L4-L2/3 synapses at somatosensory cortex (and as observed in the hippocampus at CA3-CA1 synapses), show pre- and/or post-synaptic plasticity requires further study. Interestingly, in the synapses studied here, t-LTD of LPP-GC and MPP-GC synapses have different requirements, indicating that the presynaptic expression of plasticity is fundamental for the correct functioning of brain circuits and it is possible that different presynaptic forms of t-LTD from synapses onto the same neuron (as pre- and postsynaptic forms of LTD, *Banerjee et al., 2009*; *Banerjee et al., 2014*) are regulated differently. Presynaptically expressed forms of t-LTD may in fact control the trial-to-trial reliability, and they may induce a larger change in signal-to-noise ratio than postsynaptic changes alone, as described in auditory cortex (*Froemke et al., 2013*). Broad-ranging input representations from the EC transmitted via the PP are sparse, and the DG processes this information and converts it into a less well-correlated output to become less similar and more specific (*Borzello et al., 2023*). Whether t-LTD at LPP- and MPP-GC synapses described here are involved in this important process require further research and the different cellular and molecular mechanisms involved in t-LTD at LPP- and MPP-GC synapses may account for some of the difference in their physiological influence. Notwithstanding this, the potential behavioural influence of the presynaptic forms of LTD studied here is still an emerging issue of particular interest in the near future.

## Materials and methods

### Animals and ethical approval

All animal procedures were conducted in accordance with the European Union Directive 2010/63/EU regarding the protection of animals used for scientific purposes, and they were approved by the Ethics Committee at the Universidad Pablo de Olavide and that of the Andalusian Government. C57BL/6 mice were purchased from Harlan Laboratories (Spain) and P13–21 (P, post-natal day) mice of either sex were used. Animals were kept at temperatures between 18 and 24 °C on a continuous 12 hr light/dark cycle, and at 40–60% humidity, with ad libitum access to food and water. In some experiments, dominant-negative (dn) SNARE mice (*Pascual et al., 2005*) of the same age were used. These mice were not fed with doxycycline from birth and the transgenes were continuously expressed. In these mice, the human glial fibrillary acidic protein (hGFAP) promoter mediates the specific expression of the tetracycline transactivator (tTA) in astrocytes, which in turn activates the tetracycline responsive element operator and drives the cytosolic expression of VAMP2/synaptobrevin II in these cells, along with the enhanced green fluorescence protein (eGFP). Expression of the dnSNARE transgene interferes with the formation of the SNARE complex, blocking exocytosis and impairing vesicle release by astrocytes (*Sultan et al., 2015*).

## Slice preparation

Hippocampal slices were prepared as described previously (*Andrade-Talavera et al., 2016*; *Pérez-Rodríguez et al., 2019*; *Falcón-Moya et al., 2020*). Briefly, mice were anesthetized with isoflurane (2%), decapitated, and their whole brain was removed and placed in an ice-cold ACSF solution (pH 7.2, 300 mOsm*$L^{-1}$) containing (in mM): 126 NaCl, 3 KCl, 1.25 $NaH_2PO_4$, 2 $MgSO_4$, 2 $CaCl_2$, 26 $NaHCO_3$, and 10 glucose. Slices (350 µm thick: Leica VT1000S Vibratome) were maintained oxygenated (95% $O_2$/5% $CO_2$) in the same solution for at least 1 hr before use. Experiments were carried out at 33–34°C and during the experiments; the slices were superfused continuously with the same solution indicated above.

## Electrophysiological recordings

Whole-cell patch-clamp recordings of DG GCs were obtained under visual guidance by infrared differential interference contrast microscopy. The identity of the neurons was verified by their characteristic voltage response to a current step protocol applied in current-clamp configuration using a patch-clamp amplifier (Multiclamp 700B), and acquiring the data with pCLAMP 10.2 software (Molecular Devices). Patch-clamp electrodes were pulled from borosilicate glass tubes and had a resistance of 4–7 MΩ when filled with the intracellular solution consisting in (in mM, pH 7.2–7.3, 290 mOsm $L^{-1}$): potassium gluconate, 110; HEPES, 40; NaCl, 4; ATP-Mg, 4; and GTP, 0.3. Only cells with a stable resting membrane potential below −55 mV were assessed, and the cell recordings were excluded from the analysis if the series resistance changed by >15%. All recordings were low-pass filtered at 3 kHz and acquired at 10 kHz. For plasticity experiments, excitatory postsynaptic potentials (EPSPs) were evoked alternately at 0.2 Hz in two input pathways, test and control, by using two monopolar stimulating electrodes placed 200–400 µm from the cell soma. Stimulating electrodes were situated on fibres contacting the distal and medial third of the dendritic arbour of GCs, corresponding to LPP- and MPP-GC synapses, respectively. Stimulation was adjusted (200 µs, 0.1–0.2 mA) to obtain an EPSP peak amplitude of 3–5 mV in control conditions. Pathway independence was ensured by the lack of cross-facilitation when the pathways were stimulated alternately at 50ms intervals. Plasticity was assessed through the changes in the EPSP slope, measured in its rising phase as a linear fit between time points corresponding to 25–30% and 70–75% of the peak amplitude under control conditions. Miniature responses were recorded in the presence of tetrodotoxin (TTX, 300 nM). Astrocytes were identified by their morphology under differential interference contrast microscopy, and were characterized by low membrane potential (–81±0.6 mV, n=55), low membrane resistance (20±0.3 MΩ, n=55) and passive responses (they do not show action potentials) to both negative and positive current injection.

## Plasticity protocols

After establishing a stable basal EPSP over 10 min, the test input was paired 100 times with a single postsynaptic spike. The single postsynaptic spike was evoked by a brief somatic current pulse (5ms, 0.1–0.5 pA), whereas the control pathway was left unstimulated during the pairing period. To induce t-LTD, a postsynaptic AP was evoked 18ms before the onset of the EPSP. EPSP slopes were monitored for at least 30 min after the pairing protocol, and the presynaptic stimulation frequency remained constant throughout the experiment. In some experiments, glutamate or WIN 55, 212–2 'puffs' were applied using a Picospritzer (Parker Hannifin), expulsing glutamate or WIN55,212–2 dissolved in the external solution through a micropipette over an astrocyte in proximity to the LPP-GC or MPP-GC synapses at a pressure of 10 psi for 50–200ms. In each experiment, 100 glutamate or WIN55,212–2 puffs were applied to the recording neuron at 0.2 Hz at baseline, 18ms before the onset of the EPSP, which did not affect patch clamping. The EPSP slopes were then monitored for 30 min.

## Drugs

The following agents were purchased from: Sigma-Aldrich - BAPTA, Zinc chloride, and the compounds used to prepare the ACSF and current clamp internal solutions; and Tocris Bioscience - MK-801 maleate, D-AP5, TTX, PPDA, Ro 25–6981 maleate, L-glutamic acid, MPEP, LY367385, LY341495, AM251, Orlistat (THL), Evans blue, WIN 55,212–2, FK506, Nimodipine, Thapsigargin, Ifenprodil, ruthenium red, bicuculline, SCH50911 and NBQX. All compounds were dissolved in distilled water except

for Nimodipine, Thapsigargin, PPDA, FK506, AM251 and THL, which were dissolved in dimethyl sulfoxide (DMSO). Tetanus toxin light chain was acquired from Creative Diagnostics.

## Data analysis

The data were analysed using the Clampfit 10.2 software (Molecular Devices) and the last 5 min of the recordings were used to estimate the changes in synaptic efficacy relative to the baseline. For the paired-pulse ratio (PPR) experiments, two EPSPs were evoked for 30 s at the baseline frequency, one at the beginning of the baseline recording (40ms apart) and another 30 min after the end of the pairing protocol. The PPR was expressed as the slope of the second EPSP divided by the slope of the first EPSP. A Coefficient of Variation (CV) analysis was carried out on the EPSP slopes (*Rodríguez-Moreno and Paulsen, 2008*; *Falcón-Moya et al., 2020*) and the noise-free CV of the EPSP slopes was calculated as:

$$CV = \sqrt{\frac{\sigma^2\left(EPSP\right) - \sigma^2\left(noise\right)}{EPSP\ slope}}$$

where $\sigma^2$ (EPSP) and $\sigma^2$ (noise) are the variance of the EPSP and baseline, respectively. The plot compares the variation in the mean EPSP slope (M) to the change in response variance of the EPSP slope ($1/CV^2$: see *Brock et al., 2020* for a comprehensive explanation). Graphs were prepared using SigmaPlot 14.0.

## Statistical analysis

Before applying any statistical comparison, the data were subjected to Shapiro-Wilk normality and equal variance tests, applying a confidence interval (CI) of 95%. Unpaired or paired Student's t-tests were used for comparisons between two groups. For multiple comparisons of more than two groups, one-way ANOVA was used followed by a Holm–Sidak *post-hoc* test (when necessary). All the data are expressed as the mean ± SEM and p values were considered significant when <0.05: *p<0.05, **p<0.01, ***p<0.001.

## Acknowledgements

We thank Dr Joao Oliveira for providing the dnSNARE mice. This work received support from the Spanish Agencia Estatal de Investigación and Fondo Europeo de Desarrollo Regional (FEDER, Grants PID2019-107677GB-I00 and PID2022-136597NB-I00), and from the Junta de Andalucía and Fondo Europeo de Desarrollo Regional (FEDER, Grant P20-0881) to A R-M. H C-C was supported by a Visiting Scholarship from the European Society of Neurochemistry (ESN), and I M-G was supported by a FPU Fellowship from the Spanish Ministerio de Ciencia, Innovación y Universidades and by PID2022-136597NB-I00 grant.

## Additional information

### Funding

| Funder | Grant reference number | Author |
| --- | --- | --- |
| Agencia Estatal de Investigación | PID2019-107677GB-I00 | Antonio Rodriguez-Moreno |
| Agencia Estatal de Investigación | PID2022-136597NB-I00 | Antonio Rodriguez-Moreno Irene Martínez-Gallego |
| Junta de Andalucía | P20-0881 | Antonio Rodriguez-Moreno |
| European Society of Neurochemistry | Visiting Scholarship | Heriberto Coatl-Cuaya |

| Funder | Grant reference number | Author |
|---|---|---|
| Ministerio de Ciencia, Innovación y Universidades | FPU Fellowship | Irene Martínez-Gallego |

The funders had no role in study design, data collection and interpretation, or the decision to submit the work for publication.

## Author contributions
Irene Martínez-Gallego, Heriberto Coatl-Cuaya, Data curation, Formal analysis, Investigation, Methodology; Antonio Rodriguez-Moreno, Conceptualization, Data curation, Supervision, Funding acquisition, Investigation, Methodology, Writing - original draft, Project administration, Writing - review and editing

## Author ORCIDs
Antonio Rodriguez-Moreno (iD) https://orcid.org/0000-0002-8078-6175

## Ethics
All experiments were conducted in accordance with the European Union Directive 2010/63/EU and they were approved by the Ethics Commitee at the University Pablo de Olavide and the Andalucian Government.

Reviewer #1 (Public review): https://doi.org/10.7554/eLife.98031.3.sa1
Reviewer #2 (Public review): https://doi.org/10.7554/eLife.98031.3.sa2
Reviewer #3 (Public review): https://doi.org/10.7554/eLife.98031.3.sa3
Author response https://doi.org/10.7554/eLife.98031.3.sa4

---

# Additional files

## Supplementary files
• MDAR checklist

## Data availability
All data generated or analysed during this study are included in the manuscript and supporting files.

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
