## [Editor Report · eLife Assessment]

This **valuable** study reports the existence of specific spike-timing dependent synaptic plasticity processes at two excitatory synapses of the dentate gyrus granule cells. These synapses link the entorhinal cortex and the dentate gyrus but via different circuits. With state-of-the-art patch-clamp electrophysiological analysis, the authors provide **convincing** information on the molecular mechanisms underlying these 2 forms of synaptic plasticity showing a critical role for astrocytes in both alongside some features distinctive to each pathway. These results will be of interest to neuroscientists as they uncover detailed plasticity mechanisms involving the hippocampus.

---

## [Referee Report · Reviewer #1 (Public review)]

Summary:

The study characterized the cellular and molecular mechanisms of spike timing-dependent long-term depression (t-LTD) at the synapses between excitatory afferents from lateral (LPP) and medial (MPP) perforant pathways to granule cells (GC) of the dentate gyrus (DG) in mice.

Strengths:

The electrophysiological experiments are thorough. The experiments are systematically reported and support the conclusions drawn.

This study extends current knowledge by elucidating additional plasticity mechanisms at PP-GC synapses, complementing existing literature.

Comments on the revised version:

The revised study introduces two additional approaches to confirm astrocyte involvement in t-LTD: loading astrocytes with tetanus toxin light chain to inhibit exocytosis, and using Evans blue to block vesicular glutamate uptake. These new findings further reinforce the conclusion that t-LTD relies on Ca2+-dependent glutamate exocytosis from astrocytes.

---

## [Referee Report · Reviewer #2 (Public review)]

Summary:

This work reports the existence of spike timing-dependent long-term depression (t-LTD) of excitatory synaptic strength at two synapses of the dentate gyrus granule cell, which are differently connected to the entorhinal cortex via either the lateral or medial perforant pathways (LPP or MPP, respectively). Using patch-clamp electrophysiological recording of tLTD in combination with either pharmacology or a genetically modified mouse model, they provide information on the differences in the molecular mechanism underlying this t-LTD at the two synapses.

Strengths:

The two synapses analyzed in this study have been understudied. This new data thus provides interesting new information on a plasticity process at these synapses, and the authors demonstrate subtle differences in the underlying molecular mechanisms at play. Experiments are in general well controlled and provide robust data that are properly interpreted.

The data provided to demonstrate that glutamate release from astrocytes is necessary for these plasticity mechanisms are strong. This is particularly interesting as another example of how astrocytes regulate synapse plasticity.

Weaknesses:

This work was performed at young synapses and the highlighted mechanisms are therefore pertinent to this age, as acknowledged by the authors. We currently don't know if these mechanisms are still at play at the adult synapse.

Significance:

While this is the first report of t-LTD at these synapses, this plasticity process has been mechanistically well investigated at other synapses in the hippocampus and in the cortex. Nevertheless, this new data suggests that mechanistic differences in the induction of t-LTD at these two DG synapses could contribute to the differences in the physiological influence of the LPP and MPP pathways.

---

## [Referee Report · Reviewer #3 (Public review)]

Coatl et al. investigated the mechanisms of synaptic plasticity of two important hippocampal synapses, the excitatory afferents from lateral and medial perforant pathways (LPP and MPP, respectively) of the entorhinal cortex (EC) connecting to granule cells of the hippocampal dentate gyrus (DG). They find that these two different EC-DG synaptic connections in mice show a presynaptically expressed form of long-term depression (LTD) requiring postsynaptic calcium, eCB synthesis, CB1R activation, astrocyte activity, and metabotropic glutamate receptor activation. Interestingly, LTD at MPP-GC synapses requires ionotropic NMDAR activation whereas LTD at LPP-GC synapse is NMDAR independent. Thus, they discovered two novel forms of t-LTD that require astrocytes at EC-GC synapses. Although plasticity of EC-DG granule cell (GC) synapses has been studied using classical protocols, These are the first analyses of the synaptic plasticity induced by spike timing dependent protocols at these synapses. Interestingly, the data also indicate that t-LTD at each type of synapse require different group I mGluRs, with LPP-GC synapses dependent on mGluR5 and MPP-GC t-LTD requiring mGluR1.

The authors performed a detailed analysis of the coefficient of variation of the EPSP slopes, miniature responses and different approaches (failure rate, PPRs, CV, and mEPSP frequency and amplitude analysis) they demonstrate a decrease in the probability of neurotransmitter release and a presynaptic locus for these two forms of LTD at both types of synapses. By using elegant electrophysiological experiments and taking the advantage of the conditional dominant-negative (dn) SNARE mice in which doxycycline administration blocks exocytosis and impairs vesicle release by astrocytes, they demonstrate that both LTD forms require the release of gliotransmitters from astrocytes. These data add in an interesting way to the ongoing discussion on whether LTD induced by STDP participates in refining synapses potentially weakening excitatory synapses under the control of different astrocytic networks. The conclusions of this paper are well supported by data.

---

## [Author Response]

The following is the authors’ response to the original reviews.

**Public Reviews:**

**Reviewer #1 (Public Review):**
Summary:The study characterized the cellular and molecular mechanisms of spike timing-dependent long-term depression (t-LTD) at the synapses between excitatory afferents from lateral (LPP) and medial (MPP) perforant pathways to granule cells (GC) of the dentate gyrus (DG) in mice.Strengths:The electrophysiological experiments are thorough. The experiments are systematically reported and support the conclusions drawn.This study extends current knowledge by elucidating additional plasticity mechanisms at PP-GC synapses, complementing existing literature.

We thank the reviewer for the positive assessment of our work and the constructive suggestions to improve the manuscript.

Weaknesses:To more conclusively define the pivotal role of astrocytes in modulating t-LTD at MPP and LPP GC synapses through SNARE protein-dependent glutamate release, as posited in this study, the authors could adopt additional methods, such as alternative mouse models designed to regulate SNARE-dependent exocytosis, as well as optogenetic or chemogenetic strategies for precise astrocyte manipulation during t-LTD induction. This would provide more direct evidence of the influence of astrocytic activity on synaptic plasticity.

We thank the reviewer for the suggestion. As stated in the manuscript and in figure 4, we already used two different approaches aBAPTA to interfere with astrocyte calcium signalling and dnSNARE mice (that have vesicular release impaired) to determine the involvement of astrocytes in the discovered forms of LTD, and both approaches clearly indicated the requirement of astrocytes for t-LTD. In BAPTA-treated astrocytes and in dnSNARE mice, t-LTD was prevented. Notwithstanding this, and as suggested by the reviewer, we used two additional approaches to confirm astrocyte participation. We loaded astrocytes with the light chain of the tetanus toxin (TeTxLC), which is known to block exocytosis by cleaving the vesicle-associated membrane protein, an important part of the SNARE complex (Schiavo et al., 1992, Nature 359, 832-835). In this experimental condition, we observed a clear lack of t-LTD at both (lateral and medial) pathways, thus confirming the requirement of astrocytes and the SNARE complex and vesicular release for both types of t-LTD. In addition, to gain more insight into the fact that glutamate is released by astrocytes, we blocked glutamate release from astrocytes by loading the astrocytes with Evans blue, known to interfere with glutamate uptake into vesicles as it inhibits the vesicular glutamate transporter (VGLUT). In this experimental condition, again t-LTD was prevented, indicating that t-LTD requires Ca2+dependent exocytosis of glutamate from astrocytes.

**Reviewer #2 (Public Review):**
Summary:This work reports the existence of spike timing-dependent long-term depression (t-LTD) of excitatory synaptic strength at two synapses of the dentate gyrus granule cell, which are differently connected to the entorhinal cortex via either the lateral or medial perforant pathways (LPP or MPP, respectively). Using patch-clamp electrophysiological recording of tLTD in combination with either pharmacology or a genetically modified mouse model, they provide information on the differences in the molecular mechanism underlying this t-LTD at the two synapses.Strengths:The two synapses analyzed in this study have been understudied. This new data thus provides interesting new information on a plasticity process at these synapses, and the authors demonstrate subtle differences in the underlying molecular mechanisms at play. Experiments are in general well controlled and provide robust data that are properly interpreted.

We thank the reviewer for the positive assessment of our work and the constructive suggestions to improve the manuscript.

Weaknesses:Caution should be taken in the interpretation of the results to extrapolate to adult brain as the data were obtained in P13-21 days old mice, a period during which synapses are still maturing and highly plastic.

We thank the reviewer for noticing this. In fact, our experiments were intentionally performed in young animals (P13-21), just knowing that this is a critical period of plasticity. We indicate that in the methods, results, and discussion (where we discuss that in some detail) sections.

In experiments where the drug FK506 or thapsigargin are loaded intracellularly, the concentrations used are as high as for extracellular application. Could there be an error of interpretation when stating that the targeted actors are necessarily in the post-synaptic neuron? Is it not possible for the drug to diffuse out of the cell as it is evident that it can enter the cell when applied extracellularly?

We thank the reviewer for rising this point. While it would be possible that these compounds cross the cell membranes, to do it and to pass to other cells, this would, in principle, require a relatively long time to occur. Additionally, to have any effect, the same concentration or a relatively high concentration of that we put into the pipette has to reach other cells. Furthermore, even if a compound is able to cross a cell membrane during the duration of an experiment, after this, it may be exposed to the extracellular fluid where will be diluted and most probably washed out. For all these reasons, we do not see this very plausible. Notwithstanding this, and as suggested, we have repeated the experiments using lower concentrations of thapsigargin (1 uM) and FK506 (1 uM), and have obtained the same results. These data are now included in the figure 3 and in the text.

The experiments implicating glutamate release from astrocytes in t-LTD would require additional controls to better support the conclusions made by the authors. As the data stand, it is not clear, how the authors identified astrocytes to load BAPTA and if dnSNARE expression in astrocytes does not indirectly perturb glutamate release in neurons.

We thank the reviewer for rising this point. We now indicate how astrocytes have been identified to load BAPTA. We reply to this in detail in the “Recommendations for the authors” from reviewer 2.

Significance:While this is the first report of t-LTD at these synapses, this plasticity process has been mechanistically well investigated at other synapses in the hippocampus and in the cortex. Nevertheless, this new data suggests that mechanistic differences in the induction of t-LTD at these two DG synapses could contribute to the differences in the physiological influence of the LPP and MPP pathways.
**Reviewer #3 (Public Review):**
Coatl et al. investigated the mechanisms of synaptic plasticity of two important hippocampal synapses, the excitatory afferents from lateral and medial perforant pathways (LPP and MPP, respectively) of the entorhinal cortex (EC) connecting to granule cells of the hippocampal dentate gyrus (DG). They find that these two different EC-DG synaptic connections in mice show a presynaptically expressed form of long-term depression (LTD) requiring postsynaptic calcium, eCB synthesis, CB1R activation, astrocyte activity, and metabotropic glutamate receptor activation. Interestingly, LTD at MPP-GC synapses requires ionotropic NMDAR activation whereas LTD at LPP-GC synapse is NMDAR independent. Thus, they discovered two novel forms of t-LTD that require astrocytes at EC-GC synapses. Although plasticity of EC-DG granule cell (GC) synapses has been studied using classical protocols, These are the first analysis of the synaptic plasticity induced by spike timing dependent protocols at these synapses. Interestingly, the data also indicate that t-LTD at each type of synapse require different group I mGluRs, with LPP-GC synapses dependent on mGluR5 and MPP-GC t-LTD requiring mGluR1.The authors performed a detailed analysis of the coefficient of variation of the EPSP slopes, miniature responses and different approaches (failure rate, PPRs, CV, and mEPSP frequency and amplitude analysis) they demonstrate a decrease in the probability of neurotransmitter release and a presynaptic locus for these two forms of LTD at both types of synapses. By using elegant electrophysiological experiments and taking advantage of the conditional dominant-negative (dn) SNARE mice in which doxycycline administration blocks exocytosis and impairs vesicle release by astrocytes, they demonstrate that both LTD forms require the release of gliotransmitters from astrocytes. These data add in an interesting way to the ongoing discussion on whether LTD induced by STDP participates in refining synapses potentially weakening excitatory synapses under the control of different astrocytic networks. The conclusions of this paper are mostly well supported by data, but some aspects the results must be clarified and extended.

We thank the reviewer for the positive assessment of our work and the constructive suggestions to improve the manuscript.

(1) It should be clarified whether present results are obtained with or without the functional inhibitory synapse activation. It is not clear if GABAergic synapses are blocked or not. If GABAergic synapses are not blocked authors must discuss whether the LTD of the EPSPs is due to a decrease in glutamatergic receptor activation or an increase in GABAergic receptor activation. Moreover, it should be recommended to analyze not only the EPSPs but also the EPSCs to address whether the decrease in synaptic transmission is caused by a decrease in the input resistance or by a decrease in the space constant (lambda).

We thank the reviewer for rising these points. GABAergic inhibition was not blocked in our experiments. The observed forms of t-LTD seem to be due to a decrease in glutamate release probability as indicated in the manuscript, mediated by the mechanism we uncover and describe here. To determine and clarify whether GABA receptors have any role in these forms of t-LTD, we repeated the experiments in the presence of the GABAA and GABAB receptors antagonists bicuculline and SCH50911, respectively. Blocking GABA receptors do not prevent or affect t-LTD at LPP- or MPP-GC synapses, that is still present and with a similar magnitude that controls. These results indicating that these receptors are not involved in these forms of t-LTD. These results are now included in the text in the results section (page 8) and as a new figure S1. In our experiments, no changes in input resistance or space constant were observed, and importantly, no changes were observed in the amplitude/slopes of EPSP in the control pathway that does not undergo plasticity protocol that we routinely use in our experiments.

(2) Authors show that Thapsigargin loaded in the postsynaptic neuron prevents the induction of LTD at both synapses. Analyzing the effects of blocking postsynaptic IP3Rs (Heparin in the patch pipette) and Ryanodine receptors (Ruthenium red in the patch pipette) is recommended for a deeper analysis of the mechanism implicated in the induction of this novel forms of LTD in the hippocampus.

We thank the reviewer for this suggestion. We repeated the experiments loading the postsynaptic cell with heparin and ruthenium red using the path pipette. In these experimental conditions, we observed that t-LTD was not affected by the heparin treatment (discharging a role of IP3Rs), but that it was prevented by the ruthenium red treatment (indicating the requirement of ryanodine receptors). We include now this data in the text (page 12) and in the Figure 3a, b, e, f.

(3) Authors nicely demonstrate that CB1R activation is required in these forms of LTD by blocking CB1Rs with AM251, however an interesting unanswered question is whether CB1R activation is sufficient to induce this synaptic plasticity. This reviewer suggests studying whether applying puffs of the CB1R agonist, WIN 55,212-2, could induce these forms of LTD.

We thank the reviewer for this suggestion. We repeated the experiments adding WIN55, 212-2 as suggested. The activation of CB1R by puffs of the agonist WIN 55, 212-2 to the astrocyte, directly induced LTD at both LPP- and MPP-GC synapses. We include now this data in the text (page 14) and in the Figure 3c, d, g, h.

(4) Finally, adding a last figure with a cartoon summarizing the proposed model of action in these novel forms of LTD would add a positive value and would help the reading of the manuscript, especially in those aspects related with the discussion of the results.

We thank the reviewer for the suggestion. We include now a figure showing the proposed mechanisms (Figure 5).

The extension of these results would improve the manuscript, which provides interesting results showing two novel forms of presynaptic t-LTD in the brain synapses with different action mechanisms probably implicated in the different aspects of information processing.
**Recommendations for the authors:**

**Reviewer #1 (Recommendations For The Authors):**
There are just a few aspects that could be clarified to bolster the authors' conclusions.

The author centered the conclusion of their study on the role of astrocytic activity in regulating these two forms of plasticity (see title). To strengthen the evidence that astrocytes are key regulators of t-LTD at MPP and LPP GC synapses by regulating SNARE protein-dependent glutamate release, additional complementary approaches should be considered, such as other mouse models enabling the control of SNARE-dependent exocytosis and/or optogenetic/chemogenetic tools to selectively manipulate astrocytes during the induction of t-LTD, thereby directly assessing the impact of astrocytic activity on synaptic plasticity. Implementing calcium imaging or glutamate sensors to visualize the dynamics of astrocytic calcium signaling and glutamate release during t-LTD could be also considered.

We thank the reviewer for the suggestion. As stated in the manuscript and in figure 4, we already used two different approaches aBAPTA to interfere with astrocyte calcium signalling and dnSNARE mice (that have vesicular release impaired) to determine the involvement of astrocytes in the discovered forms of LTD, and both approaches clearly indicated the requirement of astrocytes for t-LTD. In BAPTA-treated astrocytes and in dnSNARE, t-LTD was prevented. Notwithstanding this, and as suggested by the reviewer, we used two additional approaches to confirm astrocytes participation. We loaded astrocytes with the light chain of the tetanus toxin (TeTxLC), which is known to block exocytosis by cleaving the vesicle-associated membrane protein, an important part of the SNARE complex (Schiavo et al., 1992, Nature 359, 832-835). In this experimental condition, we observed a clear lack of t-LTD at both (lateral and medial) pathways, thus confirming the requirement of astrocytes and the SNARE complex and vesicular release for both types of t-LTD. In addition, to gain more insight into the fact that glutamate is released by astrocytes, we blocked glutamate release from astrocytes by loading the astrocytes with Evans blue, known to interfere with glutamate uptake into vesicles as it inhibits the vesicular glutamate transporter (VGLUT). In this experimental condition, again t-LTD was prevented, indicating that t-LTD requires Ca2+-dependent exocytosis of glutamate from astrocytes. This information is now included in the text, pages 14 and 15 and in figure 4.

How were astrocytes identified to be loaded with BAPTA? The author should clarify this methodological aspect and provide confocal images of patched astrocytes situated 50-100 um from the recorded neuron.

We thank the reviewer for the comment. We include now this information in the Methods section (page 6) and in figure S3. Astrocytes were identified by their rounded morphology under differential interference contrast microscopy, and were characterized by low membrane potential, low membrane resistance and passive responses (they do not show action potentials) to both negative and positive current injection.

Please provide confocal images of EGFP expression in the DG astrocytes of dnSNARE mice both on and off Dox, to verify transgene expression in astrocytes

We thank the reviewer for this suggestion. We now include an image of GFP expression in the DG astrocytes of off Dox dnSNARE mice. We did not provide the animals with doxycycline since birth and thus the gene was constantly expressed. We now show this image in Fig. S3. All the pups and mice are not DOX fed, meaning that the transgenes are continuously being expressed and therefore the exocytosis should be blocked in astrocytes.

Minor points:Lines 250-253: It is mentioned that TTX is added at baseline, washed out for the t-LTD experiment, and then reapplied post t-LTD. I suggest clarifying the timing and rationale for this application for a broad audience.

We thank the reviewer for the suggestion. We now include some information related to the timing and rationale of the experiment phases (page 9).

The discussion is quite detailed and provides a comprehensive overview of the study's findings. To enhance clarity and impact, the authors might consider to,add subheadings and bullet points for key findings. This will improve readability.this section could benefit from streamlining to avoid redundancy.some sentences could be made more concise without losing meaning.

We thank the reviewer for these suggestions. We now include subheadings in the discussion section to improve readability and have made some sentences more concise and simple without losing meaning.

In figure legends, consistency with capitalization should be maintained, for example in the statistical significance notation, ***P < 0.001" or ***p < 0.001"

We now include p<0.001 in the figure legend 4 for consistency.

**Reviewer #2 (Recommendations For The Authors):**
Major:All results were obtained in young still quite immature synapses. To strengthen the significance of the findings, the authors could repeat some of the main experiments in adult mice (8 weeks and beyond). If not, they should state clearly that these mechanisms were only evidenced in early post-natal conditions.

We thank the reviewer for noticing this. In fact, our experiments were intentionally performed in young animals (P13-21), just knowing that this is a critical period of plasticity. As the reviewer suggests, we indicate that in the methods (page 5), results (page 8), and discussion (page 19) (where we discuss that in some detail) sections.

Lines 246-249 and fig 1f,p: Authors need to perform a statistical test on these two graphs to support their claim that 'A plot of CV-2 versus the change in the mean evoked EPSP 246 slope (M) before and after t-LTD mainly yielded points below the diagonal line at LPP-GC and MPP-GC synapses'.

That could not be clear in the previous version. We observed an error in the points (with some points missing) of one of the graphs that we have corrected. In addition, and as suggested by the reviewer we performed a regression analysis that confirms the conclusions stated. This is now included in the text (page 9). Thus, we have added information about mean values ± SEM in the text and the linear regression of the data for LPP-GC (Mean = 0.607 ± 0.054 vs 1/CV2 = 0.439 ± 0.096, R2 = 0.337; n = 14) and MPP-GC synapses (Mean = 0.596 ± 0.056 vs 1/CV2 = 0.461 ± 0.090, R2 = 0.168; n = 13), respectively. Data yielded on the dotted horizontal line, 1/CV2 = 1, indicates no change in the probability of release, in contrast, data yielded below the dotted diagonal line is suggestive of a change in the probability of release parameters (for review, see Brock et al., 2020, Front Synaptic Neurosci 12, 11).

We are not sure that the experiment with the MK801 provided in the patch pipet can be interpreted correctly (Figure 2 a,b and e,f). How sure are the authors that, when applying MK801 in the patch pipet, it can reach its binding site within the pore? The concentration of MK801 is also very high (500 microM) and used at the same concentration extracellularly and intracellularly. Why did the authors not use lower concentration when applied intracellularly?

We thank the reviewer for rising this point. MK801 in the pipette is reaching the pore when loaded postsynaptically as when we record NMDA currents from postsynaptic neurons loaded with MK801, these currents are blocked. We include now a control experiment showing the effect of postsynaptic MK801 on NMDA current in the text (page 10). NMDA currents has been recorded at +40 mV, blocking AMPAR and GABAR with NBQX and bicuculline. Related to the concentration, it has been described that the affinity from the internal site is much lower (several orders of magnitude) than from the extracellular side(Sun et al., 2018 Neuropharmacology, 143, 122-129) and the concentrations used have been extensively used in previous studies. It is clear that the concentrations used in the present work blocked NMDAR currents but did not prevent LTD.

Linked to the point above, for the intracellular application of FK506 and thapsigargin, the concentrations used extracellularly and intracellularly are identical. The authors could have used lower concentrations for the intracellular application. Also, how can they be sure of the correct interpretation of these data as the drug essentially reaching a post-synaptic target when applied intracellularly? If the drug can enter the neuron, why could it not diffuse out of the neuron especially when loaded at a high concentration? Maybe using a lower concentration when applied intracellularly could at least partially address this issue.It is evident that it can enter the cell when applied extracellularly?

We thank the reviewer for rising this point. While it would be possible that these compound cross the cell membranes, to do it and to pass to other cells, this would, in principle, require a relatively long time to occur. Additionally, to have any effect, the same concentration or a relatively high concentration of that we put into the pipette has to reach other cells. Furthermore, even if a compound is able to cross a cell membrane during the duration of an experiment, after this, it may be exposed to the extracellular fluid where it will be diluted and most probably washed out. For all these reasons, we do not see this very plausible. Notwithstanding this, we have repeated the experiments using lower concentrations of thapsigargin (1 uM) and FK506 (1 uM) and have obtained the same results. These data are now included in the figure 3 and the numbers in the text have been updated (pages 12-13).

The data supporting the possibility of glutamate release by astrocytes as a main source of glutamate to promote t-LTD needs to be strengthened. In experiment Figure a-h, it is not clear how the authors recognize astrocytes to patch. No details are provided in the methods or in the main text. If we understand correctly, it is only by performing a current steps protocol to ensure that the patched cell did not produce action potentials. If this was the case, the authors need to be more specific and provide details of this protocol. More importantly, the one trace that was provided in Figures 4a and 4f suggests, albeit by a rough estimation that we made with a ruler, that the highest current step only depolarized the cell to about -40 mV. This is not sufficient to ensure that the recorded cell is not a neuron. The authors should increase their steps to high depolarizing currents to ensure that the patched cell is not a neuron. Better yet, they should load the cell with an dye to process the slice after the electrophysiological recording for immunohistochemistry to ensure that it was indeed an astrocyte. Alternatively, they can try to aspirate the cell content at the end of the recording to perform a qPCR for astrocyte markers eg. GFAP.

We thank the reviewer for the comment. We include now information regarding how astrocytes were identified (also raised by reviewer 1) in the Methods section (page 6) and in figure S3. Astrocytes were identified by their rounded morphology under differential interference contrast microscopy, eGFP fluorescence (astrocytes from dnSNARE mice), and were characterized by low membrane potential, low membrane resistance and passive responses (they do not show action potentials) to both negative and positive current injection.

We agree with the reviewer that in figure 4a and 4f, the step protocol might not be completely clear. For this, we revised that and now include in a clearer way that we applied pulses that depolarized astrocytes beyond -20 mV, with no action potentials found at any point. We also include now this in figure S3.

Related to the point above, the use of the model expressing dnSNARE in astrocytes is elegant. Yet, to really interpret the data obtained in these slices as a lack of vesicle release (and most importantly glutamate) we think that the authors should ensure that glutamate release from nearby neurons is not impacted. They could patch nearby neurons in dnSNARE slices and test PPR or synaptic fatigue when stimulating either the LPP or MPP. The authors should avoid overinterpretation of these results. As it stands, it is not evident that dnSNARE expression does not perturb other mechanisms within the astrocyte that in turn perturb pre-synaptic glutamate release. Adding back glutamate as puffs does not help to disentangle this issue.

To gain more insight into the fact that glutamate is released by astrocytes we blocked glutamate release from astrocytes by loading the astrocytes with Evans blue, known to interfere with glutamate uptake into vesicles as it inhibits the vesicular glutamate transporter (VGLUT). In this experimental condition, as indicated above, t-LTD was prevented, indicating that t-LTD requires Ca2+-dependent exocytosis of glutamate from astrocytes. This is included in the text (page 15) and in figure 4d,e, i, j.

In addition, we loaded astrocytes with the light chain of the tetanus toxin (TeTxLC) which is known to block exocytosis by cleaving the vesicle-associated membrane protein, an important part of the SNARE complex (Schiavo et al., 1992, Nature 359, 832-835). In this experimental condition, we observed a clear lack of t-LTD at both (lateral and medial) pathways, thus confirming the requirement of astrocytes and the SNARE complex and vesicular release for both types of t-LTD. These data indicate that t-LTD requires Ca2+-dependent exocytosis of glutamate from astrocytes. This information is now included in the text, page 14 and in figure 4.

Minor points:line 107, did the authors mean t-LTP and t-LTD? we don't understand STDP mentioned here.

We meant to say t-LTP. This is now corrected.

line 108: should STDP be replaced by t-LTD as the authors only focused on this plasticity mechanism.

We agree, we indicate now t-LTD.

line 131-132 : it is not clear when the animals were fed with doxycycline. If it was from birth, then the 'not' should be removed. Otherwise the authors should clearly state when the doxycyline was provided.

DOX was not provided and that means that the transgene was continuously expressed and therefore the exocytosis should be blocked in astrocytes. We express that clearer in page 5, methods section.

line 223 : which hippocampal synapses? needs to be stated

As suggested this is now included in the text as for cortical synapses. Synapses are Schaffer collaterals SC-CA1 for hippocampus and layer L4-L2/3 for cortical synapses (page 8).

line 273: what do the authors mean when writing 'from'? We don't understand the data provided on this line.

We thank the reviewer for noticing this. That refers to the amplitude of NMDAR-mediated currents average before and after D-AP5 or MK801. We express this now in a clearer way (page 10, from 57±8 pA to 6±5 pA).

line 286 : why do the authors point out work on GluN2B and GluN3A only here when they first investigate GluN2A contribution to t-LTD? what about previous data on GluN2A?

We have now expressed this in a different way to make it clear. We wanted to indicate that the available data for presynaptic NMDAR at MPP-GC synapses has been indicated to contain GluN2B and GluN3A subunits and to our knowledge, no data indicate that they contain GluN2A subunits.

line 428 : what do the authors mean by 'not least' ?

This is a typo and we have removed that from the text.

**Reviewer #3 (Recommendations For The Authors):**
My only suggestion for improving data presentation in the manuscript would be to split some figures of the paper. In my opinion, the figures are too dense and therefore difficult to follow for the broad audience of eLife readers. In addition, a real image of the recorded dentate granule cells in the slice showing also the location of the real stimulation electrodes would significantly improve the presentation of Figure 1.

We thank the reviewer for the suggestion, but we would prefer to let the figures as they are organized, as while we agree in some cases they are a bit big, in this way it is easier to compare lateral and medial pathways. For this, it could be better to let information regarding the two pathways in the same figure. Nevertheless, we try now to make figures clearer to use a columnar organization of the figures for each pathway what we think, would make easier to compare pathways. As the reviewer suggests we include now a real image of the recorded dentate granule cells in the slice showing also the location of the real stimulation electrodes in Figure 1, that we agree will improve the presentation of this figure and thank the reviewer for the suggestion.